# Investigating the Relationship between Work-To-Family Conflict, Job Burnout, Job Outcomes, and Affective Commitment in the Construction Industry

**DOI:** 10.3390/ijerph17165995

**Published:** 2020-08-18

**Authors:** Cong Liu, Jiming Cao, Peng Zhang, Guangdong Wu

**Affiliations:** 1School of Economics and Management, Tongji University, Shanghai 200092, China; liucong1993@tongji.edu.cn (C.L.); caojm@tongji.edu.cn (J.C.); 2School of Public Affairs, Chongqing University, Chongqing 400044, China; gd198410@cqu.edu.cn

**Keywords:** affective commitment, job burnout, job outcomes, work-to-family conflict

## Abstract

This study explored the effects of work-to-family conflict on job burnout and job outcomes in the construction industry, focusing on the moderating effects of affective commitment. Based on the conservation of resources theory, a theoretical model introducing affective commitment as a moderating variable was established. A structured questionnaire survey was then implemented among construction professionals in China. A total of 376 valid responses were obtained. Structural equation modeling was used to analyze the valid data. The results revealed the following: (i) work-to-family conflict has a significant positive impact on job burnout, but a significant negative impact on job satisfaction and job performance; (ii) job burnout negatively affects job satisfaction and job performance; (iii) affective commitment negatively moderates the effects of work-to-family conflict on job burnout. This study provides a reference for construction companies to manage work-to-family conflict and job burnout of employees, while also improving their affective commitment and job outcomes.

## 1. Introduction

The construction industry is a labor-intensive and project-oriented industry. As such, it is characterized by a heavy workload, complex processes, high risks, and long construction periods [1]. The industry is often seen as full of difficulties, challenges and dangers [2]. Practitioners in the construction industry (i.e., construction professionals) include the technicians and middle and senior managers of the owner teams, the contractor teams, the supervisor teams, the consultant teams, and the designer teams [1]. Construction professionals face the common and traditional challenges inherent in projects, including managing quality, duration and cost [3]. However, they also face increasingly urgent safety and environmental issues [1,4]. To realize smooth project delivery, many construction companies encourage employees to devote more time and energy to work, which may include sacrificing evenings, weekends and holidays [1,2]. These factors combine to result in construction professionals working long hours under tremendous and sustained pressure. These conditions may lead to job burnout [5], which involves the mental state of fatigue, cynicism and inefficiency that employees experience under long-term work stress. Job burnout is considered the main response to the work stress experienced by employees [6]. It can negatively affect individual health-related outcomes, such as mental health [7,8], and may affect job-related outcomes, including job satisfaction and job performance.

Compared with many other industries, the average work times in the construction industry are longer [3], because each construction project is temporary, occurs only once, and is unique [4]. The project implementation process involves dynamic and changing internal and external environments, and many uncertain factors [1,4]. This often leads to complex tasks and process arrangements and many challenges [2]. In addition, implementing a construction project involves rigorous node planning [4]. To accomplish tasks and realize the project plan, practitioners in the construction industry must devote significant time and energy to work [1,2]. This prevents them from effectively fulfilling family duties, including raising children, being with spouses, and caring for the elderly [3,6]. This ultimately causes work-to-family conflict (WFC), a type of inter-role conflict that reflects the contradiction of role pressure from work and family domains [1]. WFC has become an increasingly severe issue in the context of construction projects [3]. It negatively affects the quality of family life, marital relationships, and the well-being of practitioners in the construction industry. It also increases their mental stress, and leads to negative emotions, including disappointment and depression [3,9]. These reactions lead to severe job burnout, ultimately affecting job satisfaction and job performance.

Scholars have begun to investigate ways to mitigate the impact of WFC on job burnout among construction professionals [5,7]. To this end, many scholars have explored the mitigating effects of support, such as organizational support [10]. However, organizational support is inadequate to mitigate the relationship between WFC and job burnout in the construction industry. This is because of project uncertainties, challenges, and the fluid environment. This makes it difficult for construction companies to provide long-term and effective support for their employees [4]. Therefore, construction professionals need to also take the initiative to adjust their own emotions and states towards the organization and work [11]. This includes their affective commitment, which is the emotional bond between individuals and their organizations, and which include identification, emotional attachment, and employee participation [12]. Individuals with affective commitment tend to have greater enthusiasm and loyalty to their organizations, and may invest more effort into completing organizational tasks and goals, even when the work falls outside their core role [13]. Affective commitment is an important driving force and valuable work resource [14], and may help mitigate the relationship between WFC and job burnout.

Previous studies have focused on the WFC issues of construction professionals [1,2,3]. However, few studies have addressed whether affective commitment can mitigate the impact of WFC on job burnout. In addition, few studies have investigated the effects of WFC on job burnout and job outcomes, such as job satisfaction and job performance. To fill this research gap, this research introduced the conservation of resources (COR) theory, and used affective commitment as a moderating variable to establish a theoretical model to investigate the relationship between WFC, job burnout, affective commitment and job outcomes (job satisfaction and job performance). Empirical testing was conducted using structural equation modeling (SEM). The objectives of this research are as follows: (i) to investigate the relationship between WFC and job burnout; (ii) to explore whether WFC significantly impacts job outcomes; (iii) to examine whether job burnout is significantly associated with job outcomes; (iv) to determine whether affective commitment has a moderating effect on the relationship between WFC and job burnout. This research provides a reference for managing the WFC and job burnout of construction professionals, as well as improving their affective commitment and job outcomes.

## 2. Theoretical Background

### 2.1. Work-to-Family Conflict (WFC)

WFC is caused by the contradiction between work role demands and family role demands [15]. Due to construction projects’ complexities and uncertainties, practitioners in the construction industry face heavy workloads and some unforeseen circumstances [1,4]. Thus, construction professionals must work for a long time under great stress, leaving little time and energy for their personal lives [3]. According to the COR theory, individual resources are limited. The excessive investment of resources in one role will cause the resources of another role to be exhausted [16]. In addition, the COR theory proposes that individuals acquire and protect the resources they value to achieve their goals, such as obtaining more income, improving the quality of life, and increasing well-being [17]. Therefore, the COR theory mainly includes two aspects, namely, resource acquisition and resource conservation [16]. Resource acquisition refers to the act whereby individuals increase their resource reserves through active interaction with the surrounding environment [18]. Resource conservation refers to that individual resisting or withdraw from situations that threaten their resources [19]. Resources include time, energy, status, and other things that people value [20]. Since the 1990s, the COR theory has been applied to explore the relationship between the work domain and the family domain [21]. It proposes that when employees devote a lot of limited resources to work, the resources they can use to fulfill their family responsibilities will be reduced [16,17]. For employees, their time and energy are limited resources. Therefore, WFC will occur when the time and energy requirements of the work domain conflict with the time and energy requirements of the family domain. In the construction industry, the high-intensity work of construction professionals consumes their limited time and energy, leading to a lack of time and energy to perform family duties [22]. This makes WFC almost inevitable for construction professionals. WFC can be divided into time-based WFC, strain-based WFC and behavior-based WFC [1,22].

Time-based WFC occurs when time demands at work take up time needed by the family [1]. Construction project technicians and managers often need to communicate with project participants (e.g., owners, contractors, designers, supervisors, consultants) and address many issues encountered in project implementation [4]. As noted above, construction projects are complex and uncertain, with many emergencies [1,4]. As a result, practitioners in the construction industry work for a long time under great stress without enough time to fulfill their family duties. This can lead to time-based WFC. Strain-based WFC occurs when work pressures limit one’s ability to meet family demands. During construction project implementation, professionals often confront high-stress tasks, such as addressing some emergencies [2,3]. In addition, dynamic and changing project environments, and a stressful working atmosphere, may cause strain-based WFC [1]. Behavior-based WFC occurs when behaviors are effective at work, but are not expected at home [22]. In a stressful construction project context, professionals facing heavy workloads may have negative emotions, such as depression and disappointment [1,10]. These can negatively impact project task completion. Thus, construction professionals need to maintain emotional resilience, appropriately adjust their emotions, and maintain calmness at work [22]. However, their family members may expect them to be enthusiastic and warm [1]. It is often difficult to adapt to these different behavioral demands between family and work, ultimately causing behavior-based WFC.

### 2.2. Job Burnout

Job burnout stems from long-term work pressure [5]. It includes emotional exhaustion, cynicism, and low professional effectiveness [23]. Specifically, emotional exhaustion is reflected in a lack of vitality caused by the exhaustion of emotional resources. Cynicism is manifested in the alienation from work and work-related interpersonal relationships. Low professional effectiveness is reflected in negative work evaluations and dissatisfaction with work results. Previous research has indicated that job burnout relates to negative personal-related and organizational-related outcomes, such as psychological and physical health issues [23,24], as well as decreased organizational efficiency and high employee turnover [8,10]. In addition, job burnout is “contagious” and can be transmitted to other employees [1]. This can lead to a wider range of negative emotions and psychological and physical health problems, which may negatively impact other employees’ work and family lives [7]. Thus, job burnout can negatively impact personal and organizational outcomes and the development of the industry.

Job burnout is the outcome of a combination of environmental factors (e.g., inflexible schedule, tremendous stress) and individual factors (e.g., poor professional ability, poor interpersonal relationships) [5]. However, many studies on job burnout have found that work pressure, organizational pressure and social pressure are the main factors leading to job burnout [7]. In the construction industry, the high risk, uncertainties, and the unique nature of projects require practitioners to complete many difficult tasks and complex processes, and address many emergencies [4]. This puts significant pressure from the organization and society on construction professionals in the implementation of construction projects. This increases the likelihood that professionals experience job burnout. Past research has indicated that construction professionals’ job burnout is more serious compared to practitioners in many other industries [5,6,7,8].

### 2.3. Affective Commitment

Organizational commitment reflects the attitude of individuals towards organizations [25]. Organizational commitment consists of three forms: affective, normative and continuous [1]. Affective commitment is manifested in attachment to the organization. Normative commitment and continuous commitment involve the perceived responsibility and the perceived demission cost, respectively. Among the three types of commitment, affective commitment is hypothesized to have the largest effect on personal work attitude [12,14]. This is because the characteristics of affective commitment include attachment to the organization, identification with the organization’s culture, and the willingness to continue to be a member of the organization [26]. Compared with normative commitment and continuous commitment, individuals with affective commitment have more motivation to realize work successes and contribute more to their organizations [12,25]. Past research has indicated that affective commitment positively affects personal outcomes, such as work output [13,14].

Affective commitment arises from individuals’ positive perceptions of their organizations [26]. In recent years, the increasing demand for work quality and output have increased the importance of affective commitment [1]. Affective commitment can guide construction professionals to develop a positive attitude towards their organization [13]. In addition, practitioners in the construction industry with affective commitment want to stay in their organization, as they identify with the organization’s culture and atmosphere, and are willing to invest efforts in order to achieve project success [27]. Thus, affective commitment is a key factor that affects the behaviors of construction professionals. A high affective commitment can motivate construction professionals to increase work efficiency, contribute more regardless of role, and enhance organizational efficiency [13,27]. These contribute to the smooth delivery of projects and the achievement of project success.

### 2.4. Job Outcomes

Job outcomes generally refer to job performance and job satisfaction [1]. Job performance is the result of completing a task within a specified time [28]. From this perspective, it can be said that the success or failure of an organization depends on job performance of the individuals in the organization [1]. It is generally believed that job performance is a series of employees’ behaviors that can be monitored, measured and evaluated with regard to achievement at the individual level [29]. Furthermore, these behaviors are also consistent with the organization’s goals. Excellent employees’ job performance is an important factor in promoting the development of the organization [1]. In performance appraisal, there are many suitable panels acting as the appraisers. Each kind of appraiser (e.g., immediate supervisor, peer rating, committees, self-evaluation, subordinate) has different characteristics [28]. This research focused on self-evaluated job performance, that is, perceived job performance, and it is summarized from the reviews of perception and evaluation of employees as regards their own behaviors or relevant behaviors [1,28]. These behaviors will affect the achievement of organizational goals. Perceived job performance can be reflected through a systematic evaluation, and the evaluation results can be used in the organization’s human resource management [28,29]. The main reason for adopting self-evaluation in this study is the work characteristics of the construction industry. In the construction industry, most construction professionals have their own special responsibilities [5]. Therefore, their supervisors may not have the opportunity to track or observe their performance while they are working. Moreover, most construction professionals work on different projects, which means that each construction professional may be a suitable appraiser for evaluating his/her own job performance [4,9]. Therefore, the measurement of employees’ job performance in this study is based on the evaluation of construction professionals regarding their own job performance.

Job satisfaction relates to personal emotional response to the work content, environment, and output [30]. Job satisfaction is influenced by many antecedents, including employee welfare, team atmosphere, and organizational culture [31]. Past studies have indicated that job satisfaction plays a mediating role in the relationship between many variables, such as the relationship between positive affectivity and work motivation in the context of complex projects [32], and the relationship between WFC and psychological strain [33]. In addition, previous research has indicated that job satisfaction moderates the relationship between many variables in the context of a project, such as the relationship between equity and extra-role behaviors [34]. Thus, job satisfaction has a close relationship with personal and organizational outcomes in the setting of a project.

## 3. Hypotheses Development and Theoretical Model

### 3.1. Hypotheses Development

#### 3.1.1. WFC and Job Burnout

WFC is a role stressor that prevents individuals from effectively performing their family duties [1]. This will have negative effects on individuals, including a low sense of well-being and a high turnover intention [9]. The COR theory posits that employees strive to obtain and preserve resources that help achieve them their goals, including promoting their living quality and a sense of family well-being [35]. Over the past 25 years, the COR theory has gradually become one of the most cited theories in the study of organizational behavior and professional behavior [33,35]. The COR theory was proposed as a motivational theory with the basic tenet that people have the motivation to protect their existing resources and acquire new resources [15]. Resources are defined as objects, states, conditions, and other things that people value. The value of resources varies among individuals and is related to their career and personal experience [16]. For example, for construction professionals, time with their family members is considered a valuable resource, while practitioners in other industries may not value it, or may even regard it as a threat to other resources. The basic principles of the COR theory are conservation and acquisition. The COR theory emphasizes that it is more harmful for individuals to lose resources than it is helpful for them to gain the resources that they lost [35]. Therefore, when employees are unable to fulfill their family duties effectively, they are likely to suffer a loss of resources. This could include a decline in a sense of family well-being and poor marriage relations [22]. Possible loss of resources is the main cause of stress, and may cause job burnout. Faced with such a situation, employees may respond via resignation and other measures to minimize resource losses [1]. For example, Wang et al. [36] revealed that WFC leads to low work satisfaction and high turnover intention among Chinese doctors, which can lead to job burnout. Rupert et al. [37] pointed out that WFC is positively associated with job burnout in practicing psychologists. Lizano et al. [38] found that WFC has a positive relationship with job burnout among child welfare workers. Singh et al. [39] revealed that the most important antecedent of job burnout is WFC among software developers.

In the implementation of construction projects, burdensome tasks, changing project demands, and complex process arrangements can lead to construction professionals being unable to effectively perform their family duties, ultimately leading to WFC [4,9]. The loss of resources caused by WFC, such as personal time and energy, a good sense of well-being and good marital relationship quality, can lead to negative emotions and decreased work from construction professionals [5,10], ultimately causing job burnout. As a direct result, these professionals may leave their organizations and find other suitable jobs, in order to better balance work and family demands and protect their limited individual resources [1,9]. This led to the following hypothesis:

**Hypothesis 1 (H1).** 
*WFC positively impacts job burnout.*


#### 3.1.2. WFC and Job Outcomes

Job satisfaction reflects the intrinsic satisfaction of construction professionals with their working environment, processes and achievements [30]. The construction industry involves heavy work, a dynamic project context, and changing project demands [4]. This demanding working environment is likely to negatively affect employees’ family lives, leading to WFC [9,22]. According to the COR theory, a person has limited time and energy [35]. However, WFC is consistently accompanied by the loss of personal time and energy, which construction professionals need in order to fulfill their family responsibilities [10]. The lost personal time and energy leads to feelings that employees have not protected the resources that should be allocated to the family, causing negative emotions, a low sense of well-being, and low job satisfaction [1,22].

Job performance is the result of completing a task within a specified time [1]. Construction projects are high-risk, complex and uncertain [1,4]. Thus, employees in the construction industry need to spend a lot of energy and time in their work. This results in the time and energy required for work excessively occupying the time and energy required for family [9]. The COR theory indicates that when employees’ personal resources reach the lowest level they can accept, they will take confrontational measures against their organizations so that they can protect the remaining resources, accepting declines in job performance [35]. Furthermore, the massive loss of personal resources caused by WFC can lead to a negative work attitude, such as low project commitment and career commitment [9,22]. These are likely to result in low job performance. In addition, the relationship between WFC and job performance has been verified in many other work contexts. For example, Obrenovic et al. [40] pointed out that in a modern working environment, WFC has a negative impact on employees’ psychological safety and psychological well-being, thereby negatively affecting their job performance. Wijayati et al. [41] found that job performance of junior high school teachers is affected by WFC in a negative direction. Zainal et al. [42] revealed that WFC is negatively correlated with service employees’ job performance. Zain and Setiawati [43] found that WFC has a negative relationship with the job performance of medical employees. Thus, the following hypotheses were developed:

**Hypothesis 2 (H2).** 
*WFC negatively impacts job satisfaction.*


**Hypothesis 3 (H3).** 
*WFC negatively impacts job performance.*


#### 3.1.3. Job Burnout and Job Outcomes

Past research has shown a close relationship between job burnout and negative organizational and individual outcomes. At the organizational level, job burnout has a close association with the decrease of organizational productivity and efficiency [7,10]. At the individual level, job burnout is closely related to low project commitment, high turnover intention, and a low sense of well-being [1,5]. In addition, job burnout is “contagious” [24]. This means that employees’ job burnout can negatively affect their colleagues. These negative outcomes can create significant costs for individuals and organizations.

The construction industry is a project-oriented industry [4]. The goal of the construction company is to achieve the smooth delivery of construction projects. This is done by completing the project node planning on time [5]. Construction projects have high risks, many uncertain factors, and complex tasks and processes [1,9]. As described above, construction professionals face tremendous and constant work stress, leading to job burnout. Job burnout is likely to cause negative emotions, such as depression and disappointment, and low job satisfaction. This may lead to low work efficiency and effectiveness [5,7], and ultimately negatively affect job performance [10]. This led to the following hypotheses:

**Hypothesis 4 (H4).** 
*Job burnout negatively impacts job satisfaction.*


**Hypothesis 5 (H5).** 
*Job burnout negatively impacts job performance.*


#### 3.1.4. The Moderating Role of Affective Commitment

It is generally believed that WFC is closely related to job burnout [5,10]. However, affective commitment has the potential to mitigate the effects of WFC on job burnout. This is because affective commitment reduces the perceived need of employees experiencing WFC to save and protect their personal resources, such as time and energy [1]. Individuals with affective commitment feel attached to the company they work for, identify with the company’s culture and goals, and want to stay in the company [25,26]. Thus, employees with affective commitment care about and protect the interests of their organizations. They are willing to invest more efforts to achieve the organization’s goals, even if demands go beyond their role duties [27]. The COR theory holds that employees who are attached to their organizations do not consider it frustrating to have sustained work stress [35]. They also do not save personal resources due to burdensome tasks [1]. In contrast, they are willing to make more of an effort to help achieve the organization’s goals. In addition, another theory can also be applied to explain the potential of affective commitment in moderating the stress–strain relationship between WFC and job burnout, that is, the broaden-and-build theory (B&B theory). The B&B theory of positive emotions points out that positive emotions can build lasting personal resources by broadening personal thoughts and behaviors [44]. Positive emotions have long-term benefits by building intellectual, social and physical reserves that can be used to manage future threats [45]. For example, positive emotions can be applied to moderate personal status [46].

As described above, affective commitment can make employees generate positive emotions and attitudes towards their organizations. According to the B&B theory of positive emotions, these positive attitudes and emotions can help build and broaden the subjective well-being of employees, thereby alleviating WFC and the resulting job burnout [47]. Practitioners in the construction industry who feel pressured because of their work’s interference with the performance of family duties often attribute this stress to heavy workloads [9,22]. This can lead to negative work attitudes and job burnout. However, those practitioners with affective commitment may reframe their attribution’s levels [1,27]. In addition, they tend to reduce the impulse to protect personal resources, ultimately reducing job burnout [12,13]. This led to the following hypothesis:

**Hypothesis 6 (H6).** 
*Affective commitment plays a negative moderating role in the influence of WFC on job burnout.*


### 3.2. Theoretical Model

WFC can cause job burnout among practitioners in the construction industry, as they suffer a latent or actual loss of resources, including personal time and energy, a sense of well-being, and good family relationships [9,10]. This may lead to their low job satisfaction. To reduce resource losses and conserve remaining resources, construction professionals may stop working hard to preserve personal resources [35], which may cause low job performance. However, affective commitment may mitigate the relationship between WFC and job burnout, as employees with affective commitment can actively regulate their status and pressure levels, ultimately reducing their impulse to conserve personal resources [1,22]. Thus, WFC, job burnout, job outcomes and affective commitment may be closely related. Based on the proposed hypotheses, this research established the theoretical model shown in Figure 1.

## 4. Variable Measurement and Pilot Test

### 4.1. Questionnaire Design

A questionnaire was designed to measure WFC, job burnout, job satisfaction, job performance and affective commitment. The development of measurement items for variables involved three steps. The first step was to identify and cite the items that previous research has demonstrated to have high levels of reliability and validity [9]. Since the original scales were compiled in English, all measurement items were back-translated and proofread. The second step was to revise the items based on features of the construction industry [4]. The third step was to determine the items through discussions on site with specialists in the field of construction management [1].

The measurement items used to measure WFC were designed with reference to the relevant literature (Cao et al. [1]; Liu et al. [9]; Bowen and Zhang [22]). The items applied to measure job burnout were designed according to previous studies (Lingard et al. [5]; Enshassi et al. [7]; Srivastava and Dey [23]). The items used to measure affective commitment were designed with reference to the relevant literature (Kaur and Mittal [12]; Ribeiro et al. [26]; Odoardi et al. [27]). The items applied to measure job satisfaction were designed according to previous studies (Cao et al. [1]; Witt and Wilson [34]; An et al. [48]). The items used to measure job performance were designed with reference to the relevant literature (Cao et al. [1]; Saetang et al. [28]; Xiong et al. [29]). The purpose of on-site discussions with specialists in the construction industry was to revise and determine the items and ensure their applicability [4]. A total of 10 specialists from different professional teams were interviewed to gather their views on the applicability of all items. Through our three rounds of face-to-face discussions and revisions with Chinese specialists in the field of construction management, we confirmed that the measurement items in this study are consistent with the cultural context of the Chinese construction industry. Meanwhile, the specialists reached an agreement on the suitability of all items, as listed in Table 1. A five-point Likert scale was applied to measure all items [9].

### 4.2. Pilot Test

The pilot test was designed to revise the original questionnaire [1]. It was implemented in construction projects in Shanghai, Jiangxi Province and Zhejiang Province in China. The respondents included technical staff and middle and senior management personnel of different professional teams. A total of 397 questionnaires were sent out through email and express delivery. Eventually, 163 questionnaires were returned. Among the responses, 106 were valid (response rate = 27%). The principles of questionnaire purification were to identify and delete any questionnaire whose (i) answers to items were clearly not serious, (ii) items were not answered, and (iii) answers of items were contradictory [4].

Before performing a pilot test, we executed a normality test on the valid sample. The normal quantile-quantile plot (Q-Q plot) was applied to assess whether the sample conformed to the normal distribution [1]. Figure 2 shows the test results. The sample distribution of different variables is almost a straight line, including WFC, affective commitment, job burnout and job outcomes. This indicated that the valid data are normally distributed and could be further tested.

The pilot test consisted of three steps. First, we applied the reliability coefficient of the corrected-item total correlation (CITC) and Cronbach’s α to examine the items. CITC reflects the items’ reliability. We deleted the items with a CITC value lower than 0.5 [1]. We applied the Cronbach’s α to examine the internal consistency of the items, which should not be less than 0.7 [4]. The larger the Cronbach’s α, the stronger the internal consistency of the items. Second, we applied the Kaiser–Meyer–Olkin (KMO) test and the Bartlett test to evaluate whether exploratory factor analysis (EFA) could be performed. In this research, the EFA could be performed for the variable with a KMO value exceeding 0.6 [1]. The third step was the implementation of the EFA. After these tests and the purification of all items, a formal questionnaire was obtained.

### 4.3. Formal Data Collection

The method of non-probability sampling was applied to collect samples. It was suitable as the respondents were practitioners in the construction industry [4,9]. Moreover, respondents were chosen based on whether they were willing to participate in this survey, rather than at random [1]. This ensured that the respondents in this study were construction professionals and they were willing to provide the relevant information required in this survey. The survey was distributed in Shanghai, Zhejiang Provinces and Jiangxi Provinces in China. The respondents included technical staff and middle and senior management personnel of different professional teams. According to the national regulations of China, the ethical approval of the Chinese Ethics Committee is compulsory for biomedical research. Because this study was not biomedical but an organizational survey research with no vulnerable groups involved, the ethical approval was not necessary.

A total of 1350 questionnaires were sent out by email and express delivery, and 409 questionnaires were eventually recovered. After screening the recovered questionnaires, 376 questionnaires were valid (response rate = 28%). These valid surveys underwent reliability and validity tests and SEM tests. Before conducting these tests, we used the Q-Q plot to verify the valid samples’ normal distribution. The results indicated that the valid samples conformed to the normal distribution. Besides this, the sample structure is shown in Table 2.

### 4.4. Confirmatory Factor Analysis

A confirmatory factor analysis (CFA) was applied to examine the appropriateness of each variable’s items. The CFA value of each variable was determined using AMOS 21.0. The CFA generated item reliability indexes and the construct reliability (CR). Items with standardized factor loadings less than 0.6 were removed [1]. CR was applied to evaluate the consistency of each variable’s items. A CR value above 0.6 indicated good construct reliability [9]. Convergence validity was examined using the average variance extracted (AVE). The AVE value of each variable should exceed 0.5 [4]. Fitting indexes, including the ratio of the chi-square statistic to the degrees of freedom ($x^{2}/\mathrm{df}$), root mean square error of approximation (RMSEA), goodness-of-fit index (GFI), comparative fit index (CFI), adjusted goodness-of-fit index (AGFI), incremental fit index (IFI) and the normed fit index (NFI), were applied to evaluate the goodness-of-fit. The value of $x^{2}/\mathrm{df}$ should be below 3 [4]. An RMSEA value less than 0.08 was considered suitable [1]. The values of GFI, CFI, AGFI, IFI and NFI should all exceed 0.9 [9].

Table 3 shows the CFA results. Each variable’s indexes met the demands. Additionally, non-response deviations and common method deviation were tested using the chi-square method and Harman’s single-factor test, respectively [1]. The results showed that these two deviations were not serious issues, indicating that the SEM tests could be performed.

## 5. Model Testing and Results

### 5.1. Control Variables Test

Before the SEM test, we evaluated whether demographic variables affect WFC, job burnout and job outcomes [1]. SPSS 23.0 was applied to conduct this test. This research indicated that gender and marital status did not significantly affect WFC, job burnout and job outcomes (gender→WFC, 0.026, *p* > 0.05; gender→JB, −0.070, *p* > 0.05; gender→JS, 0.043, *p* > 0.05; gender→JP, −0.147, *p* > 0.05; marital status→WFC, 0.009, *p* > 0.05; marital status→JB, 0.012, *p* > 0.05; marital status→JS, 0.026, *p* > 0.05; marital status→JP, 0.031, *p* > 0.05). Additionally, considering that work experience and job position may impact WFC, job burnout and job outcomes [4,33], this research tested the effects of these two variables on WFC, job burnout and job outcomes. The results indicated that work experience and job position did not significantly affect WFC, job burnout and job outcomes (work experience→WFC, 0.003, *p* > 0.05; work experience→JB, 0.114, *p* > 0.05; work experience→JS, 0.016, *p* > 0.05; work experience→JP, 0.219, *p* > 0.05; job position→WFC, 0.017, *p* > 0.05; job position→JB, 0.203, *p* > 0.05; job position→JS, 0.135, *p* > 0.05; job position→JP, 0.116, *p* > 0.05). We also considered whether older and younger employees react differently to WFC and job burnout [22]. Therefore, we conducted a homogeneity of variance test to evaluate WFC (Levene statistic = 0.209, *p* > 0.05) and job burnout (Levene statistic = 0.158, *p* > 0.05). The results showed that the hypothesis of homogeneity of variance was valid, suggesting that elderly and young practitioners in the construction industry react similarly to WFC and job burnout.

### 5.2. SEM Test

SEM was applied to test the theoretical model in this study [4]. It has been widely applied in studies of the construction industry [1,4]. The SEM analysis was implemented using AMOS 21.0. The results are shown in Figure 3 and Table 4, and indicate that the fitting indexes meet the requirements.

Table 4 shows that all hypotheses passed the test. First, the impact of WFC on job burnout is positive (WFC→JB, 0.307, *p* < 0.001), supporting H1. Second, the impact of WFC on job satisfaction and job performance is negative (WFC→JS, −0.315, *p* < 0.05; WFC→JP, −0.272, *p* < 0.01), supporting H2 and H3. Third, the impact of job burnout on job satisfaction and job performance is negative (JB→JS, −0.216, *p* < 0.05; JB→JP, −0.163, *p* < 0.01), supporting H4 and H5. Fourth, affective commitment has a negative moderating effect on the relationship between WFC and job burnout (WFC × AC→JB, −0.124, *p* < 0.05). This supports H6, indicating that affective commitment can mitigate the impact of WFC on job burnout.

A slope analysis was conducted to advance our understanding of the moderating role of the interactions between terms. Figure 4 shows the associated slope, with the nonparallel line indicating the presence of a moderation. The green, blue and red lines represent the high, medium and low status, respectively, of the moderator. The result of the moderating effect analysis shows that affective commitment has a negative moderating impact on the relationship between WFC and job burnout.

## 6. Discussion

### 6.1. Effects of WFC on Job Burnout

The results indicate a positive relationship between WFC and job burnout. This finding was consistent with Lingard and Francis [10], and further verified that WFC can aggravate job burnout in the project context. The construction project has the characteristics of complexity, uncertainty and a long construction period [4]. This leaves construction professionals with little time for family activities. According to COR theory, if employees spend a lot of resources such as time and energy in one role, they will reduce their resource investment in another role [22]. This inhibits their ability to meet the needs of the latter role [24]. If employees do not have sufficient resources to meet the needs of the work domain and the family domain, they will face the problem of WFC [23]. Therefore, when employees have a high workload and high work intensity, they will invest a lot of time and energy in their work, and have few resources to take care of their family members [26]. This increases their likelihood of experiencing WFC. In the construction industry, the dynamic project environment and changing project demands create great pressure for construction professionals [22], making it difficult to effectively address family duties. Most practitioners in the construction industry are neither young nor single [1]. Many are both responsible for the completion of the project node planning, and are also responsible for their family duties [9]. Being unable to effectively fulfill family duties may lead to WFC and eventually job burnout. This is manifested in behavior such as unwillingness to face heavy work and a lack of enthusiasm for work. In addition, many practitioners in the construction industry also receive additional assignments by email while at home [6]. These additional tasks do not necessarily produce high work efficiency and performance, but instead may reduce job satisfaction and the sense of well-being, and increase job burnout.

### 6.2. Effects of Job Burnout on Job Outcomes

This research found that job burnout has a negative impact on job satisfaction and job performance. This further verifies the conclusion of Enshassi et al. [7] that job burnout can lead to poor job outcomes. Implementing a construction project involves many interdependent tasks and processes [4]. This requires construction professionals to have good emotional states and positive work attitudes, in order to ensure effective cooperation and to complete project tasks on schedule [22]. Nevertheless, job burnout can lead to negative attitudes towards work, including decreased enthusiasm for work and disrespect for colleagues [7,8]. Thus, job burnout can reduce trust among construction professionals, undermining their relationships, decreasing organizational cohesion, and reducing the efficiency and effectiveness of cooperation among construction professionals [5]. In addition, job burnout can lead to low job satisfaction and a decreased sense of well-being. According to the COR theory, individuals have limited resources, such as time, energy and emotional resources (e.g., well-being) [35]. When individuals lose a lot of their resources to their work, they are more likely to experience burnout, depression and poor physiological outcomes. The loss of resources has a profound negative impact on personal well-being. Therefore, individuals will take actions to avoid the loss of resources [10]. In the construction industry, when construction professionals lose a lot of time with their family members due to their work, they may enact withdrawal actions, such as absenteeism and resignation [6,10]. This will cause low job performance and directly affect the completion of the project plan, eventually negatively affecting project performance and project success.

### 6.3. Effects of WFC on Job Outcomes

This study indicated that WFC negatively impacts job satisfaction and job performance. This conclusion is consistent with the conclusion of An et al. [48] that WFC is negatively related to job outcomes in a Chinese project context. Nevertheless, this result is different from that of Allen et al. [49], who found that WFC does not have a significant relationship with job outcomes. The potential explanation is that with the development of China’s economy and society, and the improvement of people’s quality of life, people’s motivations have changed from the pursuit of high salaries to the pursuit of health and higher levels of professional welfare [1]. One’s level of professional welfare and personal health status is closely associated with job satisfaction, a sense of well-being, and job performance [30,31]. In the construction industry, features such as persistently changing project demands, complex tasks and processes, and unforeseen difficulties require practitioners to spend a lot of time in their work [2,8]. This can ultimately cause WFC. The COR theory suggests that people strive to obtain and protect resources they consider valuable. Resources include money, marital status, time and energy [22]. The COR theory points out that pressure will appear when an individual’s central resources are threatened or lost, and when an individual cannot obtain essential resources after investing a large amount of resources [35]. For practitioners in the Chinese construction industry, the time they can spend with their family members is precious [1]. Therefore, when construction professionals have little time to accompany their family members, they feel a lot of psychological pressure. The great psychological stress brought on by WFC can lead to negative emotions [22]. This can cause construction professionals’ low job satisfaction, low job performance, and increased turnover intention. The rapid development of China’s economy and construction industry provides many jobs for practitioners in the construction industry [1]. Therefore, construction professionals may choose to resign and seek other job opportunities that offer increased family welfare. This can negatively affect project task completion, eventually impacting project plans.

### 6.4. The Moderating Effect of Affective Commitment

This study’s results indicate that affective commitment plays a negative moderating role in the relationship between WFC and job burnout. This conclusion supplements the existing knowledge related to WFC, by investigating how affective commitment mitigates the relationship between WFC and job burnout. Self-justification arguments can lead construction professionals to attribute the high psychological stress brought on by WFC with the high-intensity work [1,3]. This can lead to negative work attitudes and job burnout. Nevertheless, affective commitment can mitigate the impact of WFC on job burnout. As an intrinsic motivating force, affective commitment is manifested in attachment to the company, recognition of the company’s culture, and an attitude of working hard to realize the company’s objectives [25,26]. As a result, affective commitment can make construction professionals reframe their attribution level. In addition, the COR theory posits that individuals with affective commitment are willing to devote personal resources, including time and energy, to their company, instead of preserving those resources [35]. Thus, construction professionals with affective commitment do not suffer from anxiety and depression due to work stress. This makes it possible for them to experience less job burnout. In addition, the B&B theory of positive emotions is another appropriate conceptual framework for explaining the moderating effect of affective commitment [44]. The B&B theory of positive emotions proposes that positive emotions can help broaden an individual’s thought–action reserve and build individual resilience [45]. In other words, under the influence of positive emotions, individuals have more inclusive social perceptions and more expansive behaviors [47]. Therefore, the expansion effect of positive emotions can promote the development of individual resources and put people on a positive growth path. By experiencing positive emotions, people will increase their personal resources, which in turn may bring them more lasting well-being and more positive outcomes in the future [46]. In the construction industry, the positive emotions brought about by affective commitment can make construction professionals have more inclusive and positive perceptions, thereby alleviating the relationship between WFC and job burnout.

## 7. Conclusions and Implications

### 7.1. Conclusions

In this research, we empirically evaluated the relationship among WFC, affective commitment, job burnout and job outcomes. The results show that (i) WFC significantly positively affects job burnout; (ii) job burnout is negatively related to job outcomes; (iii) WFC has a significant negative effect on job outcomes; and (iv) affective commitment plays a negative moderating role in the relationship between WFC and job burnout. These conclusions help us to understand the nonfunctional effects of WFC and job burnout, as well as the functional and moderating effects of affective commitment. In addition, this research extends the existing research into WFC and affective commitment into the construction management domain.

### 7.2. Theoretical Implications

This research explains WFC, job burnout, affective commitment and job outcomes by linking them in the project environment. First, this research expands the existing knowledge body of WFC and affective commitment, confirms the nonfunctional impacts of WFC and job burnout, and captures the interaction among WFC, job burnout and job outcomes. In addition, the research suggests that socio-economic factors need to be considered when discussing the different consequences of WFC. This helps explain the important role of socio-economic factors in WFC research.

Second, the results of this research highlight the functional effect and moderating effect of affective commitment. Construction projects are high-risk, temporary, and contain many uncertain factors and unforeseen difficulties. Construction professionals’ affective commitment is important to job outcomes and project performance, and deserves more attention. Furthermore, achieving job outcomes is an important topic in the construction management research. However, few studies have discussed the impact of personal-related variables on job outcomes. This research supplements existing studies by using WFC as an antecedent variable.

### 7.3. Practical Implications

This research has some implications for construction companies. First, the occupational health of employees is becoming an increasingly important resource for companies [1]. For employees who can bring direct economic benefits to the company, improving their occupational health and welfare is increasingly seen as a reliable investment rather than a cost [46]. This is because employees with good physical and mental health will bring more positive outcomes to the company, such as higher team performance, enhanced organizational cohesion, and a better organizational atmosphere [22]. Therefore, construction companies should attach importance to the issues of WFC and job burnout of employees, endeavor to become family-friendly companies, and establish a family-supporting organization climate, rather than just emphasizing project plans [50]. Construction companies may consider making appropriate adjustments to project assignments in order to guarantee that employees have the time needed to fulfill family responsibilities. Companies should also provide employees sufficient time to spend with their families before they join other projects. In addition, if construction professionals must work overtime, the construction company should provide proper compensation, including overtime pay and vacations [51].

Second, construction companies should cultivate affective commitment in construction professionals. This may strengthen their sense of identity and emotional attachment to the organizations, and improve work passion and individual job performance. These positive emotions and behaviors make construction professionals willing to do more positive in-role and extra-role behaviors for their organization, thereby promoting the realization of organizational goals [44]. In addition, the positive behaviors brought about by affective commitment make construction professionals more willing to look at work from a positive perspective and bring a positive working atmosphere to the organization [46]. These are the foundations of organizational success [47]. Measures to reinforce construction professionals’ affective commitment may include providing ample decision-making opportunities and more promotion opportunities, as well as applying efficient team incentive measures. Third, good communication mechanisms between individuals and their companies are necessary [1]. The active communication between employees and their organizations, and the organizations’ support for their employees, clearly distinguish the prosperous organizations from the declining organizations [9]. Successful organizations show more concern and a broader range of ideas and initiatives with regard to their employees, while unsuccessful organizations have poor communication with their employees [22]. Bidirectional communication can help construction companies better understand the specific difficulties employees face in balancing work and family, allowing them to provide corresponding support and help. These measures are likely to reduce WFC and the resulting job burnout of construction professionals. They may also reinforce their levels of affective commitment, enhancing their work enthusiasm and performance and contributing to project success.

### 7.4. Limitations and Future Work

The limitations of this research may include three aspects. First, the research subjects come from specific areas of China. Future studies could include subjects from other countries, to further study the relationship between WFC, job burnout, affective commitment and job outcomes. Second, this research used affective commitment as the moderating variable to investigate the influence of WFC on job burnout. Future studies could focus on other moderating variables, such as leadership and team atmosphere. Third, affective commitment may be complex in specific situations, highlighting the need to study the mechanisms driving affective commitment. Fourth, this study did not consider the effects of family-to-work conflict and work-family enrichment on job burnout and job outcomes among construction professionals. These two variables can be incorporated into independent variables for further research. In addition, this research lacks any objective job performance evaluation. Future research can adopt the strategy of combining perceived job performance evaluation with objective job performance evaluation in order to improve the reliability of research results.

Although this research has some limitations, the findings provide guidance for construction companies with regard to managing WFC and job burnout among construction professionals, improving their level of affective commitment, and promoting positive job outcomes and project performance.

## Figures and Tables

**Figure 1 ijerph-17-05995-f001:**
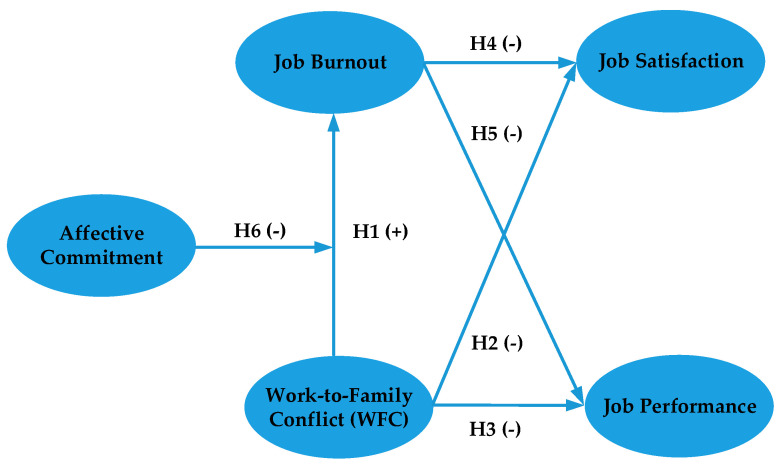
Theoretical model.

**Figure 2 ijerph-17-05995-f002:**
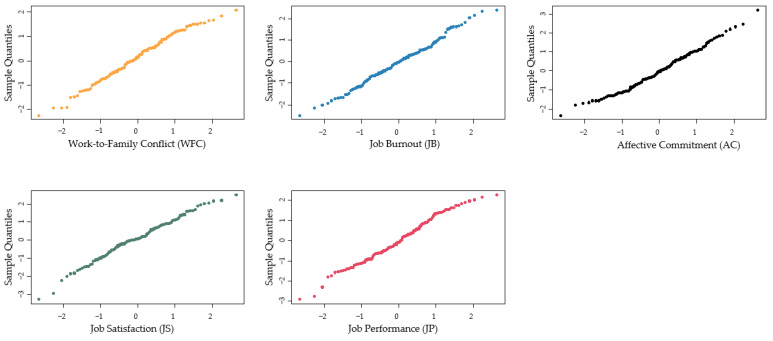
The results of the normality test.

**Figure 3 ijerph-17-05995-f003:**
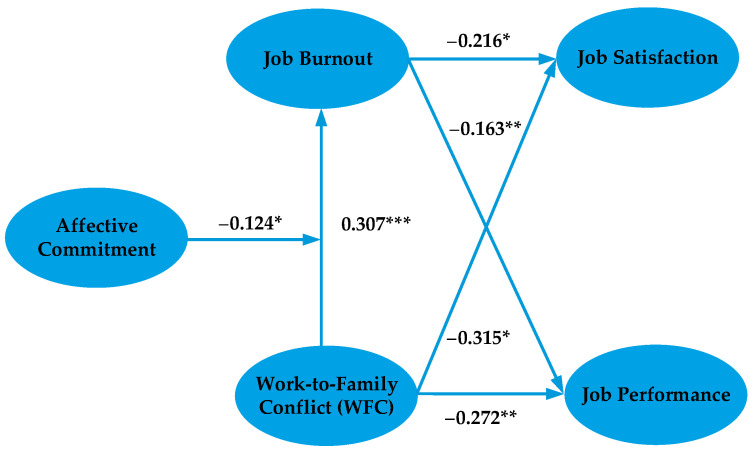
SEM test results. Note: *, *p* < 0.05. **, *p* < 0.01. ***, *p* < 0.001.

**Figure 4 ijerph-17-05995-f004:**
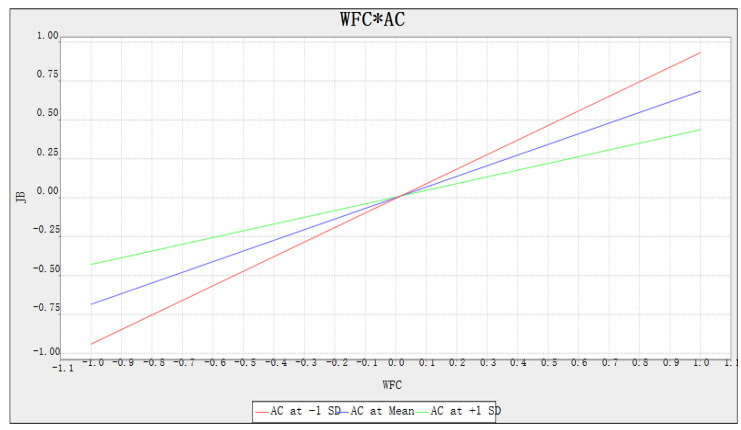
Moderating effect of affective commitment. Note: AC, affective commitment. JB, job burnout.

**Table 1 ijerph-17-05995-t001:** Measurements for variables.

Variables	No.	Measurement	References
Work-to-Family Conflict (WFC)	WFC1	The demands of work interfere with my family life.	[1,9,22]
	WFC2	I work long hours every day, so it’s hard for me to fulfill family responsibilities.	
	WFC3	Work stress makes me have a lot of negative emotions and unwilling to fulfill family responsibilities.	
	WFC4	After a whole day’s work, I am not interested in participating in family activities.	
	WFC5	Due to the project task, I often change family activity plans.	
Job Burnout (JB)	JB1	Work makes my mind and body tired.	[5,7,23]
	JB2	I feel exhausted when the work is over.	
	JB3	I feel very tired because I face many project tasks every day.	
	JB4	Working all day makes me feel stressed.	
	JB5	I began to lose interest in my work.	
	JB6	I began to doubt the meaning of work.	
Affective commitment (AC)	AC1	I like the culture and organizational atmosphere of the construction company I work for now.	[12,26,27]
	AC2	I will be proud to tell others about the construction company I work for now.	
	AC3	I want to help the construction company achieve project success.	
	AC4	I hope my efforts can help the construction company I work for achieve better development.	
	AC5	I think the construction company I’m working for is the best.	
Job Satisfaction (JS)	JS1	I like my current job and job position.	[1,34,48]
	JS2	The current leader of the company I work for is very good.	
	JS3	I like the current working environment.	
	JS4	I get along well with my colleagues.	
	JS5	I have many promotion opportunities in the company I work for.	
Job Performance (JP)	JP1	The completion of my work is excellent.	[1,28,29]
	JP2	My efforts at work have been recognized by the organization.	
	JP3	I am an excellent employee in the company.	
	JP4	The cooperation between me and my colleagues is very good.	
	JP5	I know what the owners expect.	

**Table 2 ijerph-17-05995-t002:** The sample structure of the valid surveys.

Characteristic	Category	Frequency	%
Gender	Male	279	74.20
	Female	97	25.80
Age	<30	74	19.68
	30–39	157	41.76
	40–50	96	25.53
	>50	49	13.03
Marital status	Single	119	31.65
	Married	257	68.35
Dependent children (aged 18 years or below)	Yes	226	60.11
	No	150	39.89
Elderly dependents	Yes	264	70.21
	No	112	29.79
Work experience	<5 years	83	22.07
	6–10 years	141	37.50
	11–15 years	76	20.21
	16–20 years	49	13.03
	>20 years	27	7.19
Job position	Project manager	42	11.17
	Department manager	76	20.21
	Project engineer	142	37.77
	Professional manager	105	27.93
	Others	11	2.92
Average hours worked per week	<40 h	27	7.18
	41–50 h	64	17.02
	51–60 h	156	41.49
	>60 h	129	34.31

**Table 3 ijerph-17-05995-t003:** Results of CFA.

Variables	CR	AVE	Fit Indices						
			**$x^{2}/\mathrm{df}$**	RMSEA	GFI	AGFI	NFI	IFI	CFI
Work-to-Family Conflict (WFC)	0.83	0.71	2.67	0.069	0.92	0.95	0.91	0.94	0.93
Job burnout (JB)	0.82	0.69	1.98	0.072	0.93	0.91	0.95	0.92	0.91
Affective commitment (AC)	0.78	0.67	1.79	0.067	0.91	0.93	0.92	0.96	0.94
Job satisfaction (JS)	0.74	0.65	1.89	0.064	0.94	0.91	0.93	0.92	0.90
Job performance (JP)	0.79	0.68	1.94	0.067	0.92	0.94	0.93	0.95	0.91

Note: confirmatory factor analysis (CFA), critical ratio (CR), average variance extracted (AVE), ratio of the chi-square statistic to the degrees of freedom ($x^{2}/\mathrm{df}$), root mean square error of approximation (RMSEA), goodness-of-fit index (GFI), adjusted goodness-of-fit index (AGFI), normed fit index (NFI), incremental fit index (IFI), comparative fit index (CFI).

**Table 4 ijerph-17-05995-t004:** Results of theoretical model analysis.

Hypothesis	Path Coefficient	Critical Ratio	Standard Error	T Statistics	*p* Values	Hypotheses Decision
WFC→JB	0.307 ***	11.276	0.189	6.704	0.000	H1: √
WFC→JS	−0.315 *	−2.514	0.102	−2.507	0.026	H2: √
WFC→JP	−0.272 **	−3.295	0.113	−3.208	0.007	H3: √
JB→JS	−0.216 *	−2.509	0.102	−2.523	0.039	H4: √
JB→JP	−0.163 **	−3.280	0.117	−3.136	0.003	H5: √
WFC × AC→JB	−0.124 *	−2.497	0.102	−2.519	0.037	H6: √
Fit indices (the full model)	$x^{2}/\mathrm{df}=2.17$; RMSEA = 0.063; GFI = 0.94; AGFI = 0.92; NFI = 0.95; IFI = 0.91; CFI = 0.93					

Note: *, *p* < 0.05. **, *p* < 0.01. ***, *p* < 0.001.

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
