# Peer review of "Investigating the Relationship between Work-To-Family Conflict, Job Burnout, Job Outcomes, and Affective Commitment in the Construction Industry"

_ijerph, 2020, doi:10.3390/ijerph17165995_

Round 1

Reviewer 1 Report

Explain why the author chose construction industry and not some other discipline? Why construction industry is special for research?

Author Response

Reviewer 1

A1: Explain why the author chose construction industry and not some other discipline? Why construction industry is special for research?

Response: Thanks for your comments. There are three reasons why we choose to study the construction industry rather than other disciplines. First, the major of our team members is construction project management. The research direction of our team is also construction project management. Two members of our team have more than 20 years of academic research experience in the field of construction management. The other two members have more than 15 years and more than 9 years of academic research experience in the field of construction management, respectively. Second, this research is an extended research of the project "The impact of conflicts in the construction industry and their evolution on project value appreciation" supported by the National Natural Science Foundation of China (71561009). This research and many of our previous published papers have focused on an increasingly serious phenomenon in the Chinese construction industry, that is, work-to-family conflict (WFC). We have listed our previously published papers related to WFC in the appendix to this question. Third, we do not know much about other industries such as the hotel industry and the tourism industry. Our team members also have no working experience or research experience in these industries. Therefore, we choose to continue to focus on the WFC issue in the construction industry and insist on researching this issue.

There are also three reasons why the construction industry is special for research. First, construction is a task-driven industry, characterized by high risks, heavy workloads, and long construction periods. Construction is often described as a difficult and hazardous industry and construction professionals face both common traditional challenges such as project quality, time and cost, as well as increasingly urgent safety and environmental issues. Over the past few decades, construction projects have become increasingly large, complex and integrated. To achieve project success, many construction enterprises encourage construction professionals to spend more time at work, including evenings, weekends, and holidays. These factors combine impose immense and prolonged pressure on construction professionals in implementing construction projects, possibly triggering job burnout. Job burnout is considered the main reaction and product of work pressure experienced by individuals. It has significant negative effects on individual-related outcomes such as well-being, health, and work commitment, and may affect job outcomes, such as job satisfaction and job performance.

Second, compared with other industries, construction industry involves more heavy workloads and longer average working hours. This is because construction projects are high risk and very complex, and their internal and external environments are highly dynamic and uncertain. This frequently leads to difficult tasks, complex processes, and unforeseen problems in the project implementation. Furthermore, construction involves strict node planning and individual performance appraisals. To complete node planning on time and achieve smooth project delivery, construction professionals must spend much time and energy at work. This prevents them from effectively performing family responsibilities, such as accompanying a spouse, caring for children and elderly family members. This ultimately leads to work-to-family conflict.

Third, as a form of conflict between roles, work-to-family conflict reflects the incompatibility of role stress coming from the work and family fields. Work-to-family conflict has become a serious problem in the construction industry. It negatively impacts quality of family life and the well-being of construction professionals, and increases their psychological pressure and triggers negative emotions, such as frustration, anxiety and angry. These are likely to increase their likelihood of experiencing job burnout, ultimately affecting their job outcomes.

References

  1. Yang, F.; Li, X.; Song, Z.; Li, Y.; Zhu, Y.; Asce, A.M. Job Burnout of Construction Project Managers : Considering the Role of Organizational Justice. J. Constr. Eng. Manag. 2018, 144, 04018103.
  2. Offia Ibem, E.; Anosike, M.N.; Azuh, D.E.; Mosaku, T.O. Work Stress Among Professionals in the Building Construction Industry in Nigeria. Australas. J. Constr. Econ. Build. 2011, 11, 45–57.
  3. Zhao, X.; Hwang, B.-G.; Lee, H.N. Identifying critical leadership styles of project managers for green building projects. Int. J. Constr. Manag. 2016, 16, 150–160.
  4. Bahadorestani, A.; Karlsen, J.T.; Farimani, N.M. Novel Approach to Satisfying Stakeholders in Megaprojects: Balancing Mutual Values. J. Manag. Eng. 2020, 36, 4019047.
  5. Lingard, H.; Sublet, A. The impact of job and organizational demands on marital or relationship satisfaction and conflict among Australian civil engineers. Constr. Manag. Econ. 2002, 20, 507–521.
  6. Huang, S.-L.; Li, R.-H.; Fang, S.-Y.; Tang, F.-C. Work Hours and Difficulty in Leaving Work on Time in Relation to Work-to-Family Conflict and Burnout Among Female Workers in Taiwan. Int. J. Environ. Res. Public Health 2020, 17, 605.
  7. Leiter, M.P.; Maslach, C. Areas of worklife: A structured approach to organizational predictors of job burnout. In Emotional and physiological processes and positive intervention strategies. Elsevier, Oxford, UK. 2004; pp. 91–134.
  8. Enshassi, A.; El-Rayyes, Y.; Alkilani, S. Job stress, job burnout and safety performance in the Palestinian construction industry. J. Financ. Manag. Prop. Constr. 2015, 20, 170–187.
  9. Yang, F.; Li, X.; Zhu, Y.; Li, Y.; Wu, C. Job burnout of construction project managers in China: A cross-sectional analysis. Int. J. Proj. Manag. 2017, 35, 1272–1287.
  10. Leung, M.; Shan Isabelle Chan, Y.; Dongyu, C. Structural linear relationships between job stress, burnout, physiological stress, and performance of construction project managers. Eng. Constr. Archit. Manag. 2011, 18, 312–328.
  11. Bowen, P.; Govender, R.; Edwards, P.; Cattell, K. Work-related contact, work–family conflict, psychological distress and sleep problems experienced by construction professionals: an integrated explanatory model. Constr. Manag. Econ. 2018, 36, 153–174.
  12. Turner, M.; Mariani, A. Managing the work-family interface: experience of construction project managers. Int. J. Manag. Proj. Bus. 2016, 9, 243–258.
  13. Lingard, H.C.; Francis, V.; Turner, M. The rhythms of project life: a longitudinal analysis of work hours and work–life experiences in construction. Constr. Manag. Econ. 2010, 28, 1085–1098.
  14. Liu, J.Y.; Low, S.P. Work-family conflicts experienced by project managers in the Chinese construction industry. Int. J. Proj. Manag. 2011, 29, 117–128.
  15. Greenhaus, J.H.; Beutell, N.J. Sources of conflict between work and family roles. Acad. Manag. Rev. 1985, 10, 76–88.
  16. Xia, N.; Zhong, R.; Wang, X.; Tiong, R. Cross-domain negative effect of work-family conflict on project citizenship behavior: Study on Chinese project managers. Int. J. Proj. Manag. 2018, 36, 512–524.

Appendix

The papers about the construction industry’s WFC we have published in the past:

  1. Cao, J.; Liu, C.; Zhou, Y.; Duan, K. Work-to-Family Conflict, Job Burnout, and Project Success among Construction Professionals: The Moderating Role of Affective Commitment. Int. J. Environ. Res. Public Health 2020, 17, 2902.
  2. Wu, G.; Wu, Y.; Li, H.; Dan, C. Job Burnout, Work-Family Conflict and Project Performance for Construction Professionals: The Moderating Role of Organizational Support. Int. J. Environ. Res. Public Health 2018, 15, 2869.
  3. Zheng, J.; Wu, G. Work-Family Conflict, Perceived Organizational Support and Professional Commitment: A Mediation Mechanism for Chinese Project Professionals. Int. J. Environ. Res. Public Health 2018, 15, 344.
  4. Wu, G.; Duan, K.; Zuo, J.; Yang, J.; Wen, S. System Dynamics Model and Simulation of Employee Work-Family Conflict in the Construction Industry. Int. J. Environ. Res. Public Health 2016, 13, 1059.
  5. Wu, G.; Hu, Z.; Zheng, J. Role Stress, Job Burnout, and Job Performance in Construction Project Managers: The Moderating Role of Career Calling. Int. J. Environ. Res. Public Health 2019, 16, 2394.
  6. Liu, B.; Wang, Q.; Wu, G.; Zheng, J; Li, L. How family-supportive supervisor affect Chinese construction workers’ work-family conflict and turnover intention: investigating the moderating role of work and family identity salience. Constr. Manag. Econ. 2020, 1–17.

Reviewer 2 Report

In general terms, it is a well-written manuscript in its different sections. It has a solid conceptual foundation, which adequately justifies the objectives and hypotheses of the study. The methodological part is rigorously developed. And the Discussion has a good degree of depth when analyzing the results obtained and their theoretical and practical implications. Despite these undoubted strengths, I suggest some slight modifications in order to continue any further.

Introduction
Although hypotheses are generally well supported, they can and should be elaborated further. Specifically, I recommend the following:
- Lines 187-188: The explanation provided in these lines must be analyzed more rigorously on the basis of the COR theory (this theory has been mentioned a few lines before, but is not adequately explained).
Hypothesis 1: Although the explanation provided by the authors is plausible to hypothesize the relationship between WFC and job burnout, this relationship has already been evidenced in other work contexts. In my opinion, citing any of these works can add solidity to the justification of the hypothesis.
- Hypothesis 3: Along the same lines as the previous comment, I consider that this hypothesis would gain solidity if we include any study that has already shown the relationship between WFC and job performance, even if it is in other work contexts (e.g., Obrenovic B, Jianguo D, Khudaykulov A and Khan MAS (2020) Work-Family Conflict Impact on Psychological Safety and Psychological Well-Being: A Job Performance Model. Front. Psychol. 11:475. doi: 10.3389/fpsyg.2020.00475)
- The moderating effect of affective commitment must be explained taking some reference theory. For example, the Broaden-and-Build Theory of Positive Emotions: Fredrickson B. (2004) The broaden-and-build theory of positive emotions. Philosophical Transactions of the Royal Society London B Biological Sciences 359, 1367–1377).

Questionnaire Design
- Line 283: the sources from which the instrument was built must be indicated.
- Were the items taken from instruments validated to the cultural context in which the research was carried out? Did they have to be translated (e.g., from English to Chinese)? If a translation process was carried out, it should be explained in detail.

Discussion
- Sections 6.2. and 6.3. must be analyzed taking into account the COR theory.
- While the moderating effect of affective commitment can be explained on the basis of COR theory, the broaden and built theory of positive emotions may be an even more appropriate conceptual framework.
- At both the organizational and worker levels, in my opinion the practical implications of the results of this study fit the perspective of positive organizational behavior (e.g., Bakker, A.B. and Schaufeli, W.B. (2008), “Positive organizational behavior: engaged employees in flourishing organizations”, Journal of Organizational Behavior, 29, pp. 147-54). I suggest that authors take this approach into consideration to emphasize the practical relevance of their findings..
- Some other limitation of the study should be indicated.

Author Response

Reviewer 2

B1: In general terms, it is a well-written manuscript in its different sections. It has a solid conceptual foundation, which adequately justifies the objectives and hypotheses of the study. The methodological part is rigorously developed. And the Discussion has a good degree of depth when analyzing the results obtained and their theoretical and practical implications. Despite these undoubted strengths, I suggest some slight modifications in order to continue any further.

Response: Thanks for your very positive comments.

B2: Although hypotheses are generally well supported, they can and should be elaborated further. Specifically, I recommend the following:
    - Lines 187-188: The explanation provided in these lines must be analyzed more rigorously on the basis of the COR theory (this theory has been mentioned a few lines before, but is not adequately explained).

Response: Thanks for your comments. We have supplemented a more rigorous explanation of the relationship between WFC and job burnout on the basis of the COR theory. Specifically, we have revised the section of “3.1.1. WFC and Job Burnout” as follows:

3.1.1. WFC and Job Burnout

WFC is a role stressor that prevents individuals from effectively performing their family duties [1]. This will have negative effects on individuals, including low sense of well-being and high turnover intention [9]. The COR theory posits that employees strive to obtain and preserve resources that help achieve their goals, including promoting their living quality and a sense of family well-being [35]. Over the past 25 years, the COR theory has gradually become one of the most cited theories in the study of organizational behavior and professional behavior [33,35]. The COR theory was proposed as a motivational theory with the basic tenet that people have the motivation to protect their existing resources and acquire new resources [15]. Resources are defined as objects, states, conditions, and other things that people value. The value of resources varies among individuals and is related to their career and personal experience [16]. For example, for construction professionals, time with their family members is considered a valuable resource, while practitioners in other industries may not value it, or may even regard it as a threat to other resources. The basic principles of the COR theory are conservation and acquisition. The COR theory emphasizes that it is more harmful for individuals to lose resources than it is helpful for them to gain the resources that they lost [35]. Therefore, when employees are unable to fulfill their family duties effectively, they are likely to suffer a loss of resources. This could include a decline in a sense of family well-being and poor marriage relations [22]. Possible loss of resources is the main cause of stress and may cause job burnout. Faced with such a situation, employees may respond through resignation and other measures to minimize resource losses [1].

B3: Hypothesis 1: Although the explanation provided by the authors is plausible to hypothesize the relationship between WFC and job burnout, this relationship has already been evidenced in other work contexts. In my opinion, citing any of these works can add solidity to the justification of the hypothesis.

Response: Thanks for your comments. We have supplemented the citations of the studies on the relationship between WFC and job burnout in other work contexts in the section of “3.1.1. WFC and Job Burnout” as follows:

3.1.1. WFC and Job Burnout

WFC is a role stressor that prevents individuals from effectively performing their family duties [1]. This will have negative effects on individuals, including low sense of well-being and high turnover intention [9]. The COR theory posits that employees strive to obtain and preserve resources that help achieve their goals, including promoting their living quality and a sense of family well-being [35]. Over the past 25 years, the COR theory has gradually become one of the most frequently cited theories in the study of organizational behavior and professional behavior [33,35]. The COR theory was proposed as a motivational theory with the basic tenet that people have an incentive to protect their existing resources and acquire new resources [15]. Resources are defined as objects, states, conditions, and other things that people value. The value of resources varies among individuals and is related to their career and personal experience [16]. For example, for construction professionals, time with their family members is considered a valuable resource, while practitioners in other industries may not value it, or may even regard it as a threat to other resources. The basic principles of the COR theory are conservation and acquisition. The COR theory emphasizes that it is more harmful for individuals to lose resources than it is helpful for them to gain the resources that they lost [35]. Therefore, when employees are unable to fulfill their family duties effectively, they are likely to suffer a loss of resources. This could include a decline in a sense of family well-being and poor marriage relations [22]. Possible loss of resources is the main cause of stress and may cause job burnout. Faced with such a situation, employees may respond through resignation and other measures to minimize resource losses [1]. For example, Wang et al. [36] revealed that WFC leads to low work satisfaction and high turnover intention among Chinese doctors, which can lead to job burnout. Rupert et al. [37] pointed out that WFC is positively associated with job burnout of practicing psychologists. Lizano et al. [38] found that WFC has a positive relationship with job burnout among child welfare workers. Singh et al. [39] revealed that the most important antecedent of job burnout is WFC among software developers.

Supplementary References in This Study:

  1. Wang, Y.; Liu, L.; Wang, J.; Wang, L. Work‐family conflict and burnout among Chinese doctors: the mediating role of psychological capital. J. Occup. Health 2012, 54, 232–240.
  2. Rupert, P.A.; Stevanovic, P.; Hunley, H.A. Work-family conflict and burnout among practicing psychologists. Prof. Psychol. Res. Pract. 2009, 40, 54–61.
  3. Lizano, E.L.; Hsiao, H.-Y.; Mor Barak, M.E.; Casper, L.M. Support in the workplace: buffering the deleterious effects of work–family conflict on child welfare workers’ well-being and job burnout. J. Soc. Serv. Res. 2014, 40, 178–188.
  4. Singh, P.; Suar, D.; Leiter, M.P. Antecedents, work-related consequences, and buffers of job burnout among Indian software developers. J. Leadersh. Organ. Stud. 2012, 19, 83–104.

B4: Hypothesis 3: Along the same lines as the previous comment, I consider that this hypothesis would gain solidity if we include any study that has already shown the relationship between WFC and job performance, even if it is in other work contexts (e.g., Obrenovic B, Jianguo D, Khudaykulov A and Khan MAS (2020) Work-Family Conflict Impact on Psychological Safety and Psychological Well-Being: A Job Performance Model. Front. Psychol. 11:475. doi: 10.3389/fpsyg.2020.00475).

Response: Thanks for your comments. We have read and cited the study you recommended to us. We also have supplemented the citations of the studies on the relationship between WFC and job performance in other work contexts in the section of “3.1.2. WFC and Job Outcomes” as follows:

3.1.2. WFC and Job Outcomes

Job performance is the result of completing a task within a specified time [1]. Construction projects are high-risk, complex, and uncertain [1,4]. Thus, employees in the construction industry need to spend a lot of energy and time in their work. This results in the time and energy required for work excessively occupying the time and energy required for family [9]. The COR theory indicates that when employees’ personal resources reach the lowest level they can accept, they will take confrontational measures against their organizations so that they can protect the remaining resources, accepting declines in job performance [35]. Furthermore, the massive loss of personal resources caused by WFC can lead to a negative work attitude, such as low project commitment and career commitment [9,22]. These are likely to result in low job performance. In addition, the relationship between WFC and job performance has been verified in many other work contexts. For example, Obrenovic et al. [40] pointed out that in a modern working environment, WFC has a negative impact on employees’ psychological safety and psychological well-being, thereby negatively affecting their job performance. Wijayati et al. [41] found that job performance of junior high school teachers is affected by WFC in a negative direction. Zainal et al. [42] revealed that WFC is negatively correlated with service employees’ job performance. Zain and Setiawati [43] found that WFC has a negative relationship with job performance of medical employees. Thus, the following hypotheses were developed:

Hypothesis 2 (H2). WFC negatively impacts job satisfaction.

Hypothesis 3 (H3). WFC negatively impacts job performance.

Supplementary References in This Study:

  1. Obrenovic, B.; Du Jianguo, A.K.; Khan, M.A.S. Work-family conflict impact on psychological safety and psychological well-being: A job performance model. Front. Psychol. 2020, 11, 00475.
  2. Wijayati, D.T.; Kautsar, A.; Karwanto, K. Emotional Intelligence, Work Family Conflict, and Job Satisfaction on Junior High School Teacher’s Performance. Int. J. High. Educ. 2020, 9, 179–188.
  3. Zainal, N.; Zawawi, D.; Aziz, Y.A.; Ali, M.H. Work-Family Conflict and Job Performance: Moderating Effect of Social Support among Employees in Malaysian Service Sector. Int. J. Bus. Soc. 2020, 21, 79–95.
  4. Zain, A.N.D.; Setiawati, T. Influence of Work Family Conflict and Job Satisfaction on Medical Employee Performance through Organizational Commitment. Rev. Integr. Bus. Econ. Res. 2019, 8, 1–19.

B5: The moderating effect of affective commitment must be explained taking some reference theory. For example, the Broaden-and-Build Theory of Positive Emotions: Fredrickson B. (2004) The broaden-and-build theory of positive emotions. Philosophical Transactions of the Royal Society London B Biological Sciences 359, 1367–1377).  

Response: Thanks for your comments. We have supplemented the application of the broaden-and-build theory of positive emotions to explain the moderating effect of affective commitment. Specifically, we have revised the section of “3.1.4. The Moderating Role of Affective Commitment” as follows:

3.1.4. The Moderating Role of Affective Commitment

It is generally believed that WFC is closely related to job burnout [5,10]. However, affective commitment has the potential to mitigate the effects of WFC on job burnout. This is because affective commitment reduces the perceived need of employees experiencing WFC to save and protect their personal resources, such as time and energy [1]. Individuals with affective commitment feel attached to the company they work for, identify with the company's culture and goals, and want to stay in the company [25,26]. Thus, employees with affective commitment care about and protect the interests of their organizations. They are willing to invest more efforts to achieve the organization's goals, even if demands go beyond their role duties [27]. The COR theory holds that employees who are attached to their organizations do not consider it frustrating to have sustained work stress [35]. They also do not save personal resources due to burdensome tasks [1]. In contrast, they are willing to make more efforts to help achieve the organization’s goals. In addition, another theory can also be applied to explain the potential of affective commitment in moderating the stress-strain relationship between WFC and job burnout, that is, the broaden-and-build theory (B&B theory). The B&B theory of positive emotions points out that positive emotions can build lasting personal resources by broadening personal thoughts and behaviors [44]. Positive emotions have long-term benefits by building intellectual, social and physical reserves that can be used to manage future threats [45]. For example, positive emotions can be applied to moderate personal status [46].

As described above, affective commitment can make employees generate positive emotions and attitudes towards their organizations. According to the B&B theory of positive emotions, these positive attitudes and emotions can help build and broaden the subjective well-being of employees, thereby alleviating WFC and the resulting job burnout [47]. Practitioners in the construction industry who feel pressured because their work’s interference with the performance of family duties often attribute this stress to heavy workloads [9,22]. This can lead to negative work attitudes and job burnout. However, those practitioners with affective commitment may reframe their attribution’s levels [1,27]. In addition, they tend to reduce the impulse to protect personal resources, ultimately reducing job burnout [12,13]. This led to the following hypothesis:

Hypothesis 6 (H6). Affective commitment plays a negative moderating role in the influence of WFC on job burnout.

Supplementary References in This Study:

  1. Fredrickson, B.L. The role of positive emotions in positive psychology: The broaden-and-build theory of positive emotions. Am. Psychol. 2001, 56, 218–226.
  2. Cohn, M.A.; Fredrickson, B.L. Beyond the moment, beyond the self: Shared ground between selective investment theory and the broaden-and-build theory of positive emotions. Psychol. Inq. 2006, 17, 39–44.
  3. Waugh, C.E.; Fredrickson, B.L. Nice to know you: Positive emotions, self–other overlap, and complex understanding in the formation of a new relationship. J. Posit. Psychol. 2006, 1, 93–106.
  4. Fredrickson, B.L. The broaden–and–build theory of positive emotions. Philos. Trans. R. Soc. London. Ser. B Biol. Sci. 2004, 359, 1367–1377.

B6: - Line 283: the sources from which the instrument was built must be indicated.

Response: Thanks for your comments. We have made a detailed supplementary explanation on the sources of the measurement items. We have revised the section of “4.1. Questionnaire Design” as follows:

4.1. Questionnaire Design

A questionnaire was designed to measure WFC, job burnout, job satisfaction, job performance and affective commitment. The development of measurement items for variables involves three steps. The first step was to identify and cite the items that previous research has demonstrated to have high levels of reliability and validity [9]. The second step was to revise the items based on features of the construction industry [4]. The third step was to determine the items through discussions on site with specialists in the field of construction management [1].

The measurement items used to measure WFC were designed with reference to the relevant literature (Cao et al. [1]; Liu et al. [9]; Bowen and Zhang [22]). The items applied to measure job burnout were designed according to previous studies (Lingard et al. [5]; Enshassi et al. [7]; Srivastava and Dey [23]). The items used to measure affective commitment were designed with reference to the relevant literature (Kaur and Mittal [12]; Ribeiro et al. [26]; Odoardi et al. [27]). The items applied to measure job satisfaction were designed according to previous studies (Cao et al. [1]; Witt and Wilson [34]; An et al. [48]). The items used to measure job performance were designed with reference to the relevant literature (Cao et al. [1]; Ikechukwu et al. [28]; Xiong et al. [29]). The purpose of on-site discussions with specialists in the construction industry was to revise and determine the items and ensure their applicability [4]. Ten specialists from different professional teams were interviewed to gather their views on the applicability of all items. After three face-to-face discussions, the specialists reached an agreement on the suitability of all items, listed in Table 1. A five-point Likert scale was applied to measure all items [9].

B7: - Were the items taken from instruments validated to the cultural context in which the research was carried out? Did they have to be translated (e.g., from English to Chinese)? If a translation process was carried out, it should be explained in detail.  

Response: Thanks for your comments. Through our three rounds of face-to-face discussions and revisions with Chinese experts in the field of construction management, we confirmed that the measurement items in this study are consistent with the cultural context of the Chinese construction industry. In addition, since the original scales were compiled in English, all measurement items were back-translated and proofread. According to the above statement, we have revised the section of “4.1. Questionnaire Design” as follows:

4.1. Questionnaire Design

A questionnaire was designed to measure WFC, job burnout, job satisfaction, job performance and affective commitment. The development of measurement items for variables involves three steps. The first step was to identify and cite the items that previous research has demonstrated to have high levels of reliability and validity [9]. Since the original scales were compiled in English, all measurement items were back-translated and proofread. The second step was to revise the items based on features of the construction industry [4]. The third step was to determine the items through discussions on site with specialists in the field of construction management [1].

The measurement items used to measure WFC were designed with reference to the relevant literature (Cao et al. [1]; Liu et al. [9]; Bowen and Zhang [22]). The items applied to measure job burnout were designed according to previous studies (Lingard et al. [5]; Enshassi et al. [7]; Srivastava and Dey [23]). The items used to measure affective commitment were designed with reference to the relevant literature (Kaur and Mittal [12]; Ribeiro et al. [26]; Odoardi et al. [27]). The items applied to measure job satisfaction were also designed according to previous studies (Cao et al. [1]; Witt and Wilson [34]; An et al. [48]). The items used to measure job performance were designed with reference to the relevant literature (Cao et al. [1]; Ikechukwu et al. [28]; Xiong et al. [29]). The purpose of on-site discussions with specialists in the construction industry was to revise and determine the items and ensure their applicability [4]. Ten specialists from different professional teams were interviewed to gather their views on the applicability of all items. Through our three rounds of face-to-face discussions and revisions with Chinese specialists in the field of construction management, we confirmed that the measurement items in this study are consistent with the cultural context of the Chinese construction industry. Meanwhile, the specialists reached an agreement on the suitability of all items, listed in Table 1. A five-point Likert scale was applied to measure all items [9].

B8: - Sections 6.2. and 6.3. must be analyzed taking into account the COR theory.

Response: Thanks for your comments. We have revised sections 6.2. and 6.3. of this study. The revised analysis of these two sections takes full account of the COR theory. The revised sections 6.2. and 6.3. are as follows:

6.2. Effects of Job Burnout on Job Outcomes

This research found that job burnout has a negative impact on job satisfaction and job performance. This further verifies the conclusion of Enshassi et al. [7] that job burnout can lead to poor job outcomes. Implementing a construction project involves many interdependent tasks and processes [4]. This requires construction professionals to have good emotional states and positive work attitudes to ensure effective cooperation and to complete project tasks on schedule [22]. Nevertheless, job burnout can lead to negative attitudes towards work, including decreased enthusiasm for work and disrespect for colleagues [7,8]. Thus, job burnout can reduce the trust among construction professionals, undermining their relationships, decreasing organizational cohesion, and reducing the efficiency and effectiveness of cooperation among construction professionals [5]. In addition, job burnout can lead to low job satisfaction and a decreased sense of well-being. According to the COR theory, individuals have limited resources, such as time, energy, and emotional resources (e.g. well-being) [35]. When individuals lose a lot of resources due to their work, they are more likely to experience burnout, depression, and poor physiological outcomes. The loss of resources has a profound negative impact on personal well-being. Therefore, individuals will take actions to avoid the loss of resources [10]. In the construction industry, when construction professionals lose a lot of time with their family members due to their work, they may take withdrawal actions, such as absenteeism and resignation [6,10]. This will cause their low job performance and directly affect the completion of the project plan, eventually negatively affecting project performance and project success.

6.3. Effects of WFC on Job Outcomes 

This study indicated that WFC negatively impacts job satisfaction and job performance. This conclusion is consistent with the conclusion of An et al. [48] that WFC is negatively related to job outcomes in a Chinese project context. Nevertheless, this result is different from Allen et al. [49], who found that WFC does not have a significant relationship with job outcomes. The potential explanation is that with the development of China's economy and society, and the improvement of people's quality of life, people's pursuit has changed from the pursuit of high salaries to the pursuit of health and higher levels of professional welfare [1]. One’s level of professional welfare and personal health status is closely associated with job satisfaction, a sense of well-being, and job performance [30,31]. In the construction industry, features such as persistently changing project demands, complex tasks and processes, and unforeseen difficulties require practitioners to spend a lot of time in their work [2,8]. This can ultimately cause WFC. The COR theory suggests that people strive to obtain and protect resources they consider valuable. Resources include money, marital status, time and energy [22]. The COR theory points out that pressure will appear when an individual's central resources are threatened or lost, and when an individual cannot obtain central resources after investing a large amount of resources [35]. For practitioners in the Chinese construction industry, the time they can spend with their family members is precious [1]. Therefore, when construction professionals have little time to accompany their family members, they feel a lot of psychological pressure. The great psychological stress brought by WFC can lead to negative emotions [22]. This can cause construction professionals’ low job satisfaction, low job performance, and increased turnover intention. The rapid development of China's economy and construction industry provides many jobs for practitioners in the construction industry [1]. Therefore, construction professionals may choose to resign and seek other job opportunities with increased family welfare. This can negatively affect project task completion, eventually impacting project plans.

B9: While the moderating effect of affective commitment can be explained on the basis of COR theory, the broaden and built theory of positive emotions may be an even more appropriate conceptual framework.

Response: Thanks for your comments. We have supplemented the application of the broaden-and-build theory of positive emotions to explain the moderating effect of affective commitment. We have revised the section of “6.4. The Moderating Effect of Affective Commitment” as follows:

6.4. The Moderating Effect of Affective Commitment

This study’s results indicate that affective commitment plays a negative moderating role in the relationship between WFC and job burnout. This conclusion supplements the existing knowledge related to WFC, by investigating how affective commitment mitigates the relationship between WFC and job burnout. Self-justification arguments can lead construction professionals to attribute the high psychological stress brought by WFC to the high-intensity work [1,3]. This can lead to negative work attitudes and job burnout. Nevertheless, affective commitment can mitigate the impact of WFC on job burnout. As an intrinsic motive force, affective commitment is manifested in attachment to the company, recognition of the company's culture, and an attitude of working hard to realize the company's objectives [25,26]. As a result, affective commitment can make construction professionals reframe their attribution level. The COR theory posits that individuals with affective commitment are willing to devote personal resources, including time and energy to their company, instead of preserving those resources [35]. Thus, construction professionals with affective commitment do not suffer from anxiety and depression due to work stress. This makes it possible for them to experience less job burnout. In addition, the B&B theory of positive emotions is another appropriate conceptual framework to explain the moderating effect of affective commitment [44]. The B&B theory of positive emotions proposes that positive emotions can help broaden an individual's thought-action reserve and build individual resilience [45]. In other words, under the influence of positive emotions, individuals have more inclusive social perceptions and more expansive behaviors [47]. Therefore, the expansion effect of positive emotions can promote the development of individual resources and put people on a positive growth path. By experiencing positive emotions, people will increase their personal resources, which in turn may bring them more lasting well-being and more positive outcomes in the future [46]. In the construction industry, the positive emotions brought about by affective commitment can make construction professionals have more inclusive and positive perceptions, thereby alleviating the relationship between WFC and job burnout.

B10: At both the organizational and worker levels, in my opinion the practical implications of the results of this study fit the perspective of positive organizational behavior (e.g., Bakker, A.B. and Schaufeli, W.B. (2008), “Positive organizational behavior: engaged employees in flourishing organizations”, Journal of Organizational Behavior, 29, pp. 147-54). I suggest that authors take this approach into consideration to emphasize the practical relevance of their findings.

Response: Thanks for your comments. We have carefully read the study you recommended to us. We also think that the practical implications of the results of our study fit the perspective of positive organizational behavior. We have revised the section of “7.3. Practical Implications” as follows:

7.3. Practical Implications

This research has some implications for construction companies. First, the occupational health of employees is becoming an increasingly important resource for companies [1]. For employees who can bring direct economic benefits to the company, improving their occupational health and welfare is increasingly seen as a reliable investment rather than a cost [46]. This is because employees with physical and mental health will bring more positive outcomes to the company, such as higher team performance, enhanced organizational cohesion, and better organizational atmosphere [22]. Therefore, construction companies should attach importance to the issues of WFC and job burnout of employees, endeavor to become family-friendly companies, and establish a family-supporting organization climate, rather than just emphasizing project plans. Construction companies may consider making appropriate adjustments to project assignments to guarantee that employees have the time needed to fulfill family responsibilities. Companies should also provide employees sufficient time to spend with their families before they join other projects. In addition, if construction professionals must work overtime, the construction company should provide proper compensation, including overtime pay and vacations.

Second, construction companies should cultivate affective commitment in construction professionals. This can strengthen their sense of identity and emotional attachment to the organizations, and improve work passion and individual job performance. These positive emotions and behaviors make construction professionals willing to do more positive in-role and extra-role behaviors for their organization, thereby promoting the realization of organizational goals [44]. In addition, the positive behaviors brought about by affective commitment make construction professionals willing to look at work from a positive perspective and bring a positive working atmosphere to the organization [46]. These are the foundations of organizational success [47]. Measures to reinforce construction professionals’ affective commitment may include providing ample decision-making opportunities and more promotion opportunities, as well as applying efficient team incentive measures. Third, good communication mechanisms between individuals and their companies are necessary [1]. The active communication between employees and their organizations and the organizations' support for their employees clearly distinguish the prosperous organizations from the declining organizations [9]. Successful organizations show more concern and a broader range of ideas and initiatives to their employees, while unsuccessful organizations have poor communication with their employees [22]. Bidirectional communication can help construction companies better understand the specific difficulties employees face in balancing work and family, allowing them to provide corresponding support and help. These measures are likely to reduce WFC and the resulting job burnout of construction professionals. It may also reinforce their levels of affective commitment, enhancing their work enthusiasm and performance and contributing to project success.

B11: Some other limitations of the study should be indicated.

Response: Thanks for your comments. We have supplemented some other limitations of this study. We have revised the section of “7.4. Limitations and Future Work” as follows:

7.4. Limitations and Future Work

The limitations of this research may include three aspects. First, the research subjects come from specific areas of China. Future studies could include subjects from other countries, to further study the relationship between WFC, job burnout, affective commitment, and job outcomes. Second, this research used affective commitment as the moderating variable to investigate the influence of WFC on job burnout. Future studies could focus on other moderating variables, such as leadership and team atmosphere. Third, affective commitment may be complex in specific situations, highlighting the need to study the mechanisms driving affective commitment. Fourth, this study did not consider the effects of family-to-work conflict and work-family enrichment on job burnout and job outcomes among construction professionals. These two variables can be incorporated into independent variables for further research. In addition, this research did not consider the influence of personality traits (e.g. the big-five personality traits). Future research can incorporate personality traits into the research to explore the responses of construction professionals to WFC and job burnout under the influence of different personality traits.

Although this research has some limitations, the findings provide guidance for construction companies to manage WFC and job burnout among construction professionals, improve their level of affective commitment, and promote positive job outcomes and project performance.

Reviewer 3 Report

The theoretical part of the article should be more conceptualised. Authors write that the theoretical bac ground is resources theory, but they do not reveal this idea. The idea of the research is revealed more in technical but not conceptual level.

The concept "job outcomes" is not described cleary. Moreover, the authors of research mislead the reader when they are writing that they were evaluating job performance in the research. As I understand the subjectively perceived job performance was evaluated and this is not the same like objective job performance (please read publications about these different concepts) but authors of this article are not discussing about that. Also they are not talking how such their decision (to evaluate subjectively perceived job performance) could influence the results of research in chapter "Limitations and future Work".

I have doubts according instruments used in research. For example, I disagree that statement AC1 is about affective committen. Also I think that items in job performance scale are not about the job performance. The scale "Job performance" is not evaluating the job performance. 

Marital status and other demographic factors could be important for WFC and job burnout relationship but authors are not evaluated that (please look to chapter 5.1).

Chapters "Discussion" and "Limitations and future Work" are superficial and they should be considered more deeply.

Author Response

Reviewer 3

C1: The theoretical part of the article should be more conceptualised. Authors write that the theoretical background is resources theory, but they do not reveal this idea. The idea of the research is revealed more in technical but not conceptual level.

Response: Thanks for your comments. We have supplemented the detailed explanation of the COR theory in the theoretical part of this research. Specifically, we have supplemented the theoretical background of the COR theory in the section of “2.1. Work-to-Family Conflict (WFC)” as follows: 

2.1. Work-to-Family Conflict (WFC)

WFC is caused by the contradiction between work role demands and family role demands [15]. Due to construction projects’ complexities and uncertainties, practitioners in the construction industry face heavy workloads and some unforeseen circumstances [1,4]. Thus, construction professionals must work for a long time under great stress, leaving little time and energy for their personal lives [3]. According to the COR theory, individual resources are limited. Excessive investment of resources in one role will cause the resources of another role to be exhausted [16]. In addition, the COR theory proposes that individuals acquire and protect the resources they value to achieve their goals, such as obtaining more income, improving the quality of life, and increasing well-being [17]. Therefore, the COR theory mainly includes two aspects, namely, resource acquisition and resource conservation [16]. Resource acquisition refers to that individuals increase their resource reserves through active interaction with the surrounding environment [18]. Resource conservation refers to that individuals resist or withdraw from situations that threaten their resources [19]. Resources include time, energy, status, and other things that people value [20]. Since the 1990s, the COR theory has been applied to explore the relationship between the work domain and the family domain [21]. It proposes that when employees devote a lot of limited resources to work, the resources they can use to fulfill their family responsibilities will be reduced [16,17]. For employees, their time and energy are limited resources. Therefore, WFC will occur when the time and energy requirements of the work domain conflict with the time and energy requirements of the family domain. In the construction industry, the high-intensity work of construction professionals consumes their limited time and energy, leading to a lack of time and energy to perform family duties [22]. This makes WFC almost inevitable for construction professionals. WFC can be divided into time-based WFC, strain-based WFC, and behavior-based WFC [1,22].

Supplementary References in This Study:

  1. Halbesleben, J.R.B.; Neveu, J.-P.; Paustian-Underdahl, S.C.; Westman, M. Getting to the “COR” understanding the role of resources in conservation of resources theory. J. Manage. 2014, 40, 1334–1364.
  2. Lin, S.-H.; Scott, B.A.; Matta, F.K. The dark side of transformational leader behaviors for leaders themselves: A conservation of resources perspective. Acad. Manag. J. 2019, 62, 1556–1582.
  3. Trzebiatowski, T.; del Carmen Triana, M. Family responsibility discrimination, power distance, and emotional exhaustion: When and why are there gender differences in work–life conflict? J. Bus. Ethics 2020, 162, 15–29.
  4. Guan, X.; Frenkel, S.J. Explaining supervisor–subordinate guanxi and subordinate performance through a conservation of resources lens. Hum. Relations 2019, 72, 1752–1775.
  5. Zhou, X.; Ma, J.; Dong, X. Empowering supervision and service sabotage: A moderated mediation model based on conservation of resources theory. Tour. Manag. 2018, 64, 170–187.
  6. Hobfoll, S.E. Conservation of resources: A new attempt at conceptualizing stress. Am. Psychol. 1989, 44, 513–524.

C2: The concept "job outcomes" is not described clearly. Moreover, the authors of research mislead the reader when they are writing that they were evaluating job performance in the research. As I understand the subjectively perceived job performance was evaluated and this is not the same like objective job performance (please read publications about these different concepts) but authors of this article are not discussing about that. Also they are not talking how such their decision (to evaluate subjectively perceived job performance) could influence the results of research in chapter "Limitations and future Work".

Response: Thanks for your comments. In the empirical study of the structural equation modeling of the construction industry, the job performance of employees refers to their perceived job performance. Since the structured questionnaire survey used in the structural equation modeling research can only measure the perceived job performance of construction professionals, we generally do not need to add the term "subjectively perceived". The same is true for the measurement and description of employees’ job performance in empirical studies of the structural equation modeling of other industries, such as the hotel industry (Choi and Kim 2012), the tourism industry (Karatepe and Kilic 2007), and the marketing industry (Babin and Boles 1998). In addition, the evaluation of job performance in this study referred to many previous relevant studies in the field of construction management (Cao et al. 2020; Ikechukwu et al. 2019; Xiong et al. 2019). In the empirical research of the structural equation modeling of the construction industry, the measurement of the job performance of construction professionals is based on their psychological level. This method of measuring employees’ job performance has been widely used in empirical research in the field of construction management. Similarly, in other industries such as the hotel industry, the tourism industry and the marketing industry, the measurement of employees’ job performance is also based on their psychological level (Choi and Kim 2012; Karatepe and Kilic 2007; Babin and Boles 1998). Despite the above statement, in order to make readers more clear about the evaluation of construction professionals’ job performance in this study, We have revised the section of “2.4. Job Outcomes” as follows:

2.4. Job Outcomes

The job outcomes generally refer to job performance and job satisfaction [1]. Job performance is the result of completing a task within a specified time [28]. Past studies have indicated that high-performing employees tend to have more career opportunities compared to low-performing people, and are more likely to be promoted in the organization [1]. In general, construction professionals’ job performance includes three dimensions: task-oriented, situational, and adaptive. Task-oriented performance relates to task-related behavior and activities, and refers to the completion of project tasks [28]. Situational performance relates to the wide range of project-related behaviors of construction professionals [1,28]. Adaptive performance is associated with employees' adaptation to changes in project demands and the project environment [29]. Adaptability involves creatively resolving project problems, addressing unforeseen emergencies, and learning new construction techniques [1]. Since this study adopted the method of the structured questionnaire survey to measure the job performance of construction professionals, the measurement of job performance in the questionnaire is based on the psychological evaluation of construction professionals on their job performance [1,2].

References

  1. Cao, J.; Liu, C.; Wu, G.; Zhao, X.; Jiang, Z. Work-family conflict and job outcomes for construction professionals: The mediating role of affective organizational commitment. Int. J. Environ. Res. Public Health 2020, 17, 1443.
  2. Ikechukwu, N.P.; Hart, R.I.; Ezeh, J.I.N.; Bridget, I.; Jude-Peters, A. Employee motivation and job performance of selected construction companies in rivers state. Int. J. Eng. Manag. Res. 2019, 9, 130–137.
  3. Xiong, B.; Newton, S.; Skitmore, M. Towards a conceptual model of the job performance of construction professionals: A person-environment fit perspective. Int. J. Constr. Manag. 2019, 1–15.
  4. Choi, H.J.; Kim, Y.T. Work‐family conflict, work‐family facilitation, and job outcomes in the Korean hotel industry. Int. J. Contemp. Hosp. Manag. 2012, 24, 1011–1028.
  5. Karatepe, O.M.; Kilic, H. Relationships of supervisor support and conflicts in the work–family interface with the selected job outcomes of frontline employees. Tour. Manag. 2007, 28, 238–252.
  6. Babin, B.J.; Boles, J.S. Employee behavior in a service environment: A model and test of potential differences between men and women. J. Mark. 1998, 62, 77–91.
  7. Wu, G.; Hu, Z.; Zheng, J. Role Stress, Job Burnout, and Job Performance in Construction Project Managers: The Moderating Role of Career Calling. Int. J. Environ. Res. Public Health 2019, 16, 2394.
  8. Wu, G.; Wu, Y.; Li, H.; Dan, C. Job Burnout, Work-Family Conflict and Project Performance for Construction Professionals: The Moderating Role of Organizational Support. Int. J. Environ. Res. Public Health 2018, 15, 2869.
  9. Wu, G.; Hu, Z.; Zheng, J. Role Stress, Job Burnout, and Job Performance in Construction Project Managers: The Moderating Role of Career Calling. Int. J. Environ. Res. Public Health 2019, 16, 2394.

C3: I have doubts according instruments used in research. For example, I disagree that statement AC1 is about affective commitment. Also I think that items in job performance scale are not about the job performance. The scale "Job performance" is not evaluating the job performance.

Response: Thanks for your comments. All measurement items in this study were designed based on relevant previous studies. We identified and cited the measurement items that have been proved by prior studies to possess high-level reliability and validity. Specifically, the measurement items used to measure WFC were designed with reference to the relevant literature (Cao et al. (2020); Liu et al. (2020); Bowen and Zhang (2020)). The items applied to measure job burnout were designed according to previous studies (Lingard et al. (2007); Enshassi et al. (2016); Srivastava and Dey (2020)). The items used to measure affective commitment were designed with reference to the relevant literature (Kaur and Mittal (2020); Ribeiro et al. (2020); Odoardi et al. (2019)). The items applied to measure job satisfaction were also designed according to previous studies (Cao et al. (2020); Witt and Wilson (1991); An et al. (2020)). The items used to measure job performance were designed with reference to the relevant literature (Cao et al. (2020); Ikechukwu et al. (2019); Xiong et al. (2019)). In addition, ten specialists from different professional teams in China's construction industry were interviewed to gather their professional opinions on the applicability of all items. Through our three rounds of face-to-face discussions and revisions with Chinese specialists in the field of construction management, we confirmed that the measurement items in this study are consistent with the cultural context of the Chinese construction industry. Meanwhile, the specialists reached an agreement on the suitability of all measurement items.

References

  1. Cao, J.; Liu, C.; Wu, G.; Zhao, X.; Jiang, Z. Work-family conflict and job outcomes for construction professionals: The mediating role of affective organizational commitment. Int. J. Environ. Res. Public Health 2020, 17, 1443.
  2. Liu, B.; Wang, Q.; Wu, G.; Zheng, J.; Li, L. How family-supportive supervisor affect Chinese construction workers’ work-family conflict and turnover intention: investigating the moderating role of work and family identity salience. Constr. Manag. Econ. 2020, 1–17.
  3. Bowen, P.; Zhang, R.P. Cross-Boundary Contact, Work-family conflict, antecedents, and consequences: Testing an integrated model for construction professionals. J. Constr. Eng. Manag. 2020, 146, 4020005.
  4. Lingard, H.C.; Yip, B.; Rowlinson, S.; Kvan, T. The experience of burnout among future construction professionals: A cross‐national study. Constr. Manag. Econ. 2007, 25, 345–357.
  5. Enshassi, A.; Al Swaity, E.; Arain, F. Investigating common causes of burnout in the construction industry. Int. J. Constr. Proj. Manag. 2016, 8, 43–56.
  6. Srivastava, S.; Dey, B. Workplace bullying and job burnout. Int. J. Organ. Anal. 2020, 28, 183–204.
  7. Kaur, P.; Mittal, A. Meaningfulness of Work and Employee Engagement: The Role of Affective Commitment. Open Psychol. J. 2020, 13, 115–122.
  8. Ribeiro, N.; Duarte, A.P.; Filipe, R.; Torres de Oliveira, R. How authentic leadership promotes individual creativity: The mediating role of affective commitment. J. Leadersh. Organ. Stud. 2020, 27, 189–202.
  9. Odoardi, C.; Battistelli, A.; Montani, F.; Peiró, J.M. Affective commitment, participative leadership, and employee innovation: a multilevel investigation. J. Work Organ. Psychol. 2019, 35, 103–113.
    10.  Witt, L.A.; Wilson, J.W. Moderating effect of job satisfaction on the relationship between equity and extra-role behaviors. J. Soc. Psychol. 1991, 131, 247–252.
  10. An, J.; Liu, Y.; Sun, Y.; Liu, C. Impact of work-family conflict, job stress and job satisfaction on seafarer performance. Int. J. Environ. Res. Public Health 2020, 17, 2191.
  11. Ikechukwu, N.P.; Hart, R.I.; Ezeh, J.I.N.; Bridget, I.; Jude-Peters, A. Employee motivation and job performance of selected construction companies in rivers state. Int. J. Eng. Manag. Res. 2019, 9, 130–137.
  12. Xiong, B.; Newton, S.; Skitmore, M. Towards a conceptual model of the job performance of construction professionals: A person-environment fit perspective. Int. J. Constr. Manag. 2019, 1–15.

C4: Marital status and other demographic factors could be important for WFC and job burnout relationship but authors are not evaluated that (please look to chapter 5.1).

Response: Thanks for your comments. We have supplemented the tests of the effects of marital status and other demographic factors on WFC and job burnout. We have revised the section of “5.1. Control Variables Test” as follows:

5.1. Control Variables Test

Before SEM test, we evaluated whether demographic variables affect WFC, job burnout, and job outcomes [1]. SPSS 23.0 was applied to conduct this test. This research indicated that gender and marital status did not significantly affect WFC, job burnout, and job outcomes (genderWFC, 0.026, p > 0.05; genderJB, -0.070, p > 0.05; genderJS, 0.043, p > 0.05; genderJP, −0.147, p > 0.05; marital statusWFC, 0.009, p > 0.05; marital statusJB, 0.012, p > 0.05; marital statusJS, 0.026, p > 0.05; marital statusJP, 0.031, p > 0.05). Additionally, considering that work experience and job position may impact WFC, job burnout, and job outcomes [4,33], this research tested the effects of these two variables on WFC, job burnout, and job outcomes. The results indicated that work experience and job position did not significantly affect WFC, job burnout, and job outcomes (work experienceWFC, 0.003, p > 0.05; work experienceJB, 0.114, p > 0.05; work experienceJS, 0.016, p > 0.05; work experienceJP, 0.219, p > 0.05; job positionWFC, 0.017, p > 0.05; job positionJB, 0.203, p > 0.05; job positionJS, 0.135, p > 0.05; job positionJP, 0.116, p > 0.05). We also considered whether older and younger employees react differently to WFC and job burnout [22]. Therefore, we conducted a homogeneity of variance test to evaluate WFC (Levene statistic = 0.209, p > 0.05) and job burnout (Levene statistic = 0.158, p > 0.05). The results showed that the hypothesis of homogeneity of variance was valid, suggesting that elderly and young practitioners in the construction industry react similarly to WFC and job burnout.

C5: Chapters "Discussion" and "Limitations and future Work" are superficial and they should be considered more deeply.

Response: Thanks for your comments. We have revised the sections “6. Discussion” and "7.4. Limitations and Future Work" as follows:

  1. Discussion

6.1. Effects of WFC on Job Burnout

The results indicate a positive relationship between WFC and job burnout. This finding was consistent with Lingard and Francis [10] and further verified that WFC can aggravate job burnout in the project context. The construction project has the characteristics of complexity, uncertainty and long construction period [4]. This leaves construction professionals with little time for family activities. According to COR theory, if employees spend a lot of resources such as time and energy in one role, they will reduce the resource investment in another role [22]. This inhibits their ability to meet the needs of the latter role [24]. If employees do not have sufficient resources to meet the needs of the work domain and the family domain, they will face the problem of WFC [23]. Therefore, when employees have a lot of workload and high work intensity, they will invest a lot of time and energy in their work and have few resources to take care of their family members [26]. This increases their likelihood of experiencing WFC. In the construction industry, the dynamic project environment and changing project demands create great pressure for construction professionals [22], making it difficult to effectively address family duties. Most practitioners in the construction industry are neither young nor single [1]. Many are both responsible for the completion of the project node planning, and are also responsible for their family duties [9]. Being unable to effectively fulfill family duties may lead to WFC and eventually job burnout. This is manifested in behavior such as unwillingness to face heavy work and lack of enthusiasm for work. In addition, many practitioners in the construction industry also receive additional assignments by email while at home [6]. These additional tasks do not necessarily produce high work efficiency and performance, but instead may reduce job satisfaction and the sense of well-being, and increase job burnout.

6.2. Effects of Job Burnout on Job Outcomes

This research found that job burnout has a negative impact on job satisfaction and job performance. This further verifies the conclusion of Enshassi et al. [7] that job burnout can lead to poor job outcomes. Implementing a construction project involves many interdependent tasks and processes [4]. This requires construction professionals to have good emotional states and positive work attitudes to ensure effective cooperation and to complete project tasks on schedule [22]. Nevertheless, job burnout can lead to negative attitudes towards work, including decreased enthusiasm for work and disrespect for colleagues [7,8]. Thus, job burnout can reduce the trust among construction professionals, undermining their relationships, decreasing organizational cohesion, and reducing the efficiency and effectiveness of cooperation among construction professionals [5]. In addition, job burnout can lead to low job satisfaction and a decreased sense of well-being. According to the COR theory, individuals have limited resources, such as time, energy, and emotional resources (e.g. well-being) [35]. When individuals lose a lot of resources due to their work, they are more likely to experience burnout, depression, and poor physiological outcomes. The loss of resources has a profound negative impact on personal well-being. Therefore, individuals will take actions to avoid the loss of resources [10]. In the construction industry, when construction professionals lose a lot of time with their family members due to their work, they may take withdrawal actions, such as absenteeism and resignation [6,10]. This will cause their low job performance and directly affect the completion of the project plan, eventually negatively affecting project performance and project success.

6.3. Effects of WFC on Job Outcomes 

This study indicated that WFC negatively impacts job satisfaction and job performance. This conclusion is consistent with the conclusion of An et al. [48] that WFC is negatively related to job outcomes in a Chinese project context. Nevertheless, this result is different from Allen et al. [49], who found that WFC does not have a significant relationship with job outcomes. The potential explanation is that with the development of China's economy and society, and the improvement of people's quality of life, people's pursuit has changed from the pursuit of high salaries to the pursuit of health and higher levels of professional welfare [1]. One’s level of professional welfare and personal health status is closely associated with job satisfaction, a sense of well-being, and job performance [30,31]. In the construction industry, features such as persistently changing project demands, complex tasks and processes, and unforeseen difficulties require practitioners to spend a lot of time in their work [2,8]. This can ultimately cause WFC. The COR theory suggests that people strive to obtain and protect resources they consider valuable. Resources include money, marital status, time and energy [22]. The COR theory points out that pressure will appear when an individual's central resources are threatened or lost, and when an individual cannot obtain central resources after investing a large amount of resources [35]. For practitioners in the Chinese construction industry, the time they can spend with their family members is precious [1]. Therefore, when construction professionals have little time to accompany their family members, they feel a lot of psychological pressure. The great psychological stress brought by WFC can lead to negative emotions [22]. This can cause construction professionals’ low job satisfaction, low job performance, and increased turnover intention. The rapid development of China's economy and construction industry provides many jobs for practitioners in the construction industry [1]. Therefore, construction professionals may choose to resign and seek other job opportunities with increased family welfare. This can negatively affect project task completion, eventually impacting project plans.

6.4. The Moderating Effect of Affective Commitment

This study’s results indicate that affective commitment plays a negative moderating role in the relationship between WFC and job burnout. This conclusion supplements the existing knowledge related to WFC, by investigating how affective commitment mitigates the relationship between WFC and job burnout. Self-justification arguments can lead construction professionals to attribute the high psychological stress brought by WFC to the high-intensity work [1,3]. This can lead to negative work attitudes and job burnout. Nevertheless, affective commitment can mitigate the impact of WFC on job burnout. As an intrinsic motive force, affective commitment is manifested in attachment to the company, recognition of the company's culture, and an attitude of working hard to realize the company's objectives [25,26]. As a result, affective commitment can make construction professionals reframe their attribution level. The COR theory posits that individuals with affective commitment are willing to devote personal resources, including time and energy to their company, instead of preserving those resources [35]. Thus, construction professionals with affective commitment do not suffer from anxiety and depression due to work stress. This makes it possible for them to experience less job burnout. In addition, the B&B theory of positive emotions is another appropriate conceptual framework to explain the moderating effect of affective commitment [44]. The B&B theory of positive emotions proposes that positive emotions can help broaden an individual's thought-action reserve and build individual resilience [45]. In other words, under the influence of positive emotions, individuals have more inclusive social perceptions and more expansive behaviors [47]. Therefore, the expansion effect of positive emotions can promote the development of individual resources and put people on a positive growth path. By experiencing positive emotions, people will increase their personal resources, which in turn may bring them more lasting well-being and more positive outcomes in the future [46]. In the construction industry, the positive emotions brought about by affective commitment can make construction professionals have more inclusive and positive perceptions, thereby alleviating the relationship between WFC and job burnout.

7.4. Limitations and Future Work

The limitations of this research may include three aspects. First, the research subjects come from specific areas of China. Future studies could include subjects from other countries, to further study the relationship between WFC, job burnout, affective commitment, and job outcomes. Second, this research used affective commitment as the moderating variable to investigate the influence of WFC on job burnout. Future studies could focus on other moderating variables, such as leadership and team atmosphere. Third, affective commitment may be complex in specific situations, highlighting the need to study the mechanisms driving affective commitment. Fourth, this study did not consider the effects of family-to-work conflict and work-family enrichment on job burnout and job outcomes among construction professionals. These two variables can be incorporated into independent variables for further research. In addition, this research did not consider the influence of personality traits (e.g. the big-five personality traits). Future research can incorporate personality traits into the research to explore the responses of construction professionals to WFC and job burnout under the influence of different personality traits.

Although this research has some limitations, the findings provide guidance for construction companies to manage WFC and job burnout among construction professionals, improve their level of affective commitment, and promote positive job outcomes and project performance. 

Round 2

Reviewer 3 Report

The answer to my remarks about methodological mistakes in the research and publication did not convince me. I still think that you should revised all the methodological part in your article (not only to add one new sentence in chapter 2.4). Once again: the concept perceived job performance differ from concept job performance. You can find research where these two mentioned constructs were measured and the results showed that the correlation between them is very low.

Author Response

Reviewer 1

A1: The answer to my remarks about methodological mistakes in the research and publication did not convince me. I still think that you should revised all the methodological part in your article (not only to add one new sentence in chapter 2.4). Once again: the concept perceived job performance differ from concept job performance. You can find research where these two mentioned constructs were measured and the results showed that the correlation between them is very low.

Response: Thanks for your comments. In this round of revisions, we have referenced a lot of literature related to perceived job performance. We have emphasized that the concept of job performance in this study is perceived job performance. We have also thoroughly revised the section of “2.4. Job Outcomes” as follows:

2.4. Job Outcomes

   The job outcomes generally refer to job performance and job satisfaction [1]. Job performance is the result of completing a task within a specified time [28]. From this perspective, it can be said that the success or failure of an organization depends on job performance of the individuals in the organization [1]. It is generally believed that job performance is a series of employees' behaviors that can be monitored, measured, and evaluated achievement in individual level [29]. Furthermore, these behaviors are also consistent with the organization's goals. Excellent employees' job performance is an important factor to promote the development of the organization [1]. In performance appraisal, there are many suitable panels being the appraisers. Each kind of appraisers (e.g., immediate supervisor, peer rating, committees, self-evaluation, subordinate) has different characteristics [28]. This research focused on self-evaluated job performance, that is, perceived job performance. It is summarized from the reviews of perception and evaluation of employees on their own behaviors or relevant behaviors [1,28]. These behaviors will affect the achievement of organizational goals. Perceived job performance can be reflected through a systematic evaluation, and the evaluation results can be used in the organization's human resource management [28,29]. The main reason for adopting self-evaluation in this study is the work characteristics of the construction industry. In the construction industry, most construction professionals have their own special responsibilities [5]. Therefore, their supervisors may not have the opportunity to track or observe their performance while they are working. Moreover, most construction professionals work on different projects, which means that each construction professional may be a suitable appraiser to evaluate his/her own job performance [4,9]. Therefore, the measurement of employees' job performance in this study is based on the evaluation of construction professionals on their own job performance.

       Job satisfaction relates to personal emotional response to the work content, environment, and output [30]. Job satisfaction is influenced by many antecedents, including employee welfare, team atmosphere, and organizational culture [31]. Past studies have indicated that job satisfaction plays a mediating role in the relationship between many variables, such as the relationship between positive affectivity and work motivation in the context of complex projects [32], and the relationship between WFC and psychological strain [33]. In addition, previous research has indicated that job satisfaction moderates the relationship between many variables in the context of a project, such as the relationship between equity and extra-role behaviors [34]. Thus, job satisfaction has a close relationship with personal and organizational outcomes in the setting of a project.

References

  1. Saetang, J.; Sulumnad, K.; Thampitak, P.; Sungkaew, T. Factors affecting perceived job performance among staff: A case study of Ban Karuna Juvenile Vocational Training Centre for boys. J. Behav. Sci. 2010, 5, 33–45.
  2. Rowold, J.; Borgmann, L.; Bormann, K. Which leadership constructs are important for predicting job satisfaction, affective commitment, and perceived job performance in profit versus nonprofit organizations? Nonprofit Manag. Leadersh. 2014, 25, 147–164.
  3. Veloutsou, C.A.; Panigyrakis, G.G. Consumer brand managers’ job stress, job satisfaction, perceived performance and intention to leave. J. Mark. Manag. 2004, 20, 105–131.
  4. Chung, S.; Lee, K.Y.; Choi, J. Exploring digital creativity in the workspace: The role of enterprise mobile applications on perceived job performance and creativity. Comput. Human Behav. 2015, 42, 93–109.
  5. Lawrence, R.; Marian, S. Women corrections officers in men’s prisons: Acceptance and perceived job performance. Women Crim. Justice 1998, 9, 63–86.
  6. Hwang, S.J.; Quast, L.N.; Center, B.A.; Chung, C.-T.N.; Hahn, H.-J.; Wohkittel, J. The impact of leadership behaviours on leaders’ perceived job performance across cultures: Comparing the role of charismatic, directive, participative, and supportive leadership behaviours in the US and four Confucian Asian countries. Hum. Resour. Dev. Int. 2015, 18, 259–277.
  7. Kalyani, L.D. An empirical investigation of the impact of organizational factors on the perceived job performance of shop floor employees of large scale garment industries in Sri Lanka. Sabaragamuwa Univ. J. 2006, 6, 82–92.
  8. Chikazhe, L.; Makanyeza, C.; Kakava, N.Z. The effect of perceived service quality, satisfaction and loyalty on perceived job performance: perceptions of university graduates. J. Mark. High. Educ. 2020, 1–18.
  9. Gaziel, H.H. Managerial studies and perceived job performance: An Israeli case study. Public Pers. Manage. 1994, 23, 341–356.
  10. Cao, J.; Liu, C.; Wu, G.; Zhao, X.; Jiang, Z. Work-family conflict and job outcomes for construction professionals: The mediating role of affective organizational commitment. Int. J. Environ. Res. Public Health 2020, 17, 1443.
  11. Ikechukwu, N.P.; Hart, R.I.; Ezeh, J.I.N.; Bridget, I.; Jude-Peters, A. Employee motivation and job performance of selected construction companies in rivers state. Int. J. Eng. Manag. Res. 2019, 9, 130–137.
  12. Choi, H.J.; Kim, Y.T. Work‐family conflict, work‐family facilitation, and job outcomes in the Korean hotel industry. Int. J. Contemp. Hosp. Manag. 2012, 24, 1011–1028.
  13. Karatepe, O.M.; Kilic, H. Relationships of supervisor support and conflicts in the work–family interface with the selected job outcomes of frontline employees. Tour. Manag. 2007, 28, 238–252.
  14. Wu, G.; Hu, Z.; Zheng, J. Role Stress, Job Burnout, and Job Performance in Construction Project Managers: The Moderating Role of Career Calling. Int. J. Environ. Res. Public Health 2019, 16, 2394.
  15. Xiong, B.; Newton, S.; Skitmore, M. Towards a conceptual model of the job performance of construction professionals: A person-environment fit perspective. Int. J. Constr. Manag. 2019, 1–13.

    After-note: the authors hope that they have now addressed all the concerns, feedback and suggestions of the Editor and Reviewers adequately. Nevertheless, if the Editor and Reviewers have any additional comments, the authors would be pleased to address these further. Thank you again for your positive contributions to make this a better paper.
